# *Plasmodium* sporozoite search strategy to locate hotspots of blood vessel invasion

Pauline Formaglio[1,6], Marina E. Wosniack[2,6], Raphael M. Tromer[3], Jaderson G. Polli [4], Yuri B. Matos[4], Hang Zhong[1], Ernesto P. Raposo[5], Marcos G. E. da Luz[4] ✉ & Rogerio Amino [1] ✉

*Plasmodium* sporozoites actively migrate in the dermis and enter blood vessels to infect the liver. Despite their importance for malaria infection, little is known about these cutaneous processes. We combine intravital imaging in a rodent malaria model and statistical methods to unveil the parasite strategy to reach the bloodstream. We determine that sporozoites display a high-motility mode with a superdiffusive Lévy-like pattern known to optimize the location of scarce targets. When encountering blood vessels, sporozoites frequently switch to a subdiffusive low-motility behavior associated with probing for intravasation hotspots, marked by the presence of pericytes. Hence, sporozoites present anomalous diffusive motility, alternating between superdiffusive tissue exploration and subdiffusive local vessel exploitation, thus optimizing the sequential tasks of seeking blood vessels and pericyte-associated sites of privileged intravasation.

Finding infrequent targets in a vast search space is a common task in nature[1–3]. For instance, to disseminate and infect the liver, *Plasmodium* sporozoites deposited in the extravascular regions of the skin during a mosquito bite first need to find and invade blood vessels (BV)[4,5], which only represent approximately 5% of the dermis volume[6,7]. After inoculation into the skin, sporozoites are known to engage in a non-Brownian movement[8], decreasing their speed upon interaction with BV[5,8]. This, nevertheless, does not necessarily result in intravasation[8], and only about 20% of inoculated sporozoites eventually gain access to the bloodstream to initiate malaria infection in the liver[5,8].

Enhanced diffusion strategies are known to maximize the likelihood of finding scarce targets[9]. Marine animals, for example, adopt a superdiffusive foraging behavior in sea regions with a low concentration of resources[10]. Numerical works also show that in patchy landscapes, more diffusive strategies boost the detection of resources[11,12]. Lévy walks are commonly used to describe such superdiffusive foraging behavior[1,2]. They essentially portray patterns of movements composed of clusters of many short steps interspersed by long steps, which can reduce oversampling and optimize intermittent search processes[13].

Here, by imaging sporozoites in vivo in a rodent malaria model and applying a Hidden Markov Model (HMM) and anomalous diffusion methods to study their motile and BV invasive behavior in the host skin, we show the strategy sporozoites use to find and intravasate at hotspots in the cutaneous microvasculature.

## Results and discussion

### Low-motility mode identified by HMM characterizes BV interaction and invasion

To study the strategy by which sporozoites find and invade BV, we continuously imaged GFP-expressing rodent-infecting *Plasmodium berghei* sporozoites within a volume of $320 \times 320 \times 30 \ \mu m^3$ during the first 40 min following their microinjection in the dermis of mouse ear[14]. Individual parasites were tracked using MTrackJ[15] and categorized either as invaders (INV) if they successfully invaded a cutaneous BV during acquisition or as non-invaders (NINV) otherwise. INV ($n = 27$)

[1]Institut Pasteur, Université Paris Cité, Malaria Infection and Immunity Unit, 75015 Paris, France. [2]Max Planck Institute for Brain Research, 60438 Frankfurt, Germany. [3]Departamento de Física Teórica e Experimental, Universidade Federal do Rio Grande do Norte, 59078- 970 Natal-RN, Brazil. [4]Departamento de Física, Universidade Federal do Paraná, 81531-980 Curitiba-PR, Brazil. [5]Laboratório de Física Teórica e Computacional, Departamento de Física, Universidade Federal de Pernambuco, 50670-901 Recife-PE, Brazil. [6]These authors contributed equally: Pauline Formaglio, Marina E. Wosniack. ✉e-mail: luz@fisica.ufpr.br; roti@pasteur.fr

were tracked on average during $422 \pm 315$ s, and NINV ($n = 87$) during $537 \pm 366$ s. Since sporozoites present behavior alternation, switching between high-speed avascular motility and slower and more constrained motility at the vicinity of BV[5,8] (Supplementary Movie 1), we used a two-state HMM[16] to analyze the data. A HMM hypothesizes that the observable states of the system being modeled, which can be described by empirical measurements, evolve according to an underlying unobservable hidden stochastic process. Moreover, the transition from one state to the next is assumed to depend only on the current state and a set of fixed probabilities, with no influence from the history of the system. Here, our observables are the step lengths ($\ell$) and turning angles ($\theta$), characteristic of the individual parasite trajectories. The hidden processes can be regarded as the behavioral (inner) state of the individual, expected in the present

case to reflect the reported high-motility and low-motility modes of sporozoites.

The trajectories of INV and NINV were analyzed using the moveHMM package in $R$[17]. Two hidden states were identified independently of the invasive phenotype (Fig. 1a). HMM state 1 (St1) revealed a low-speed motility state (Fig. 1b), with predominant turning angles around 0 and $\pm\pi$ radians (Fig. 1c) representing parasites frequently moving on or around BV (Fig. 1a, d). Conversely, HMM state 2 (St2) mainly revealed sporozoites moving farther from vessels (Fig. 1a, d), displaying a high-speed motility state (Fig. 1b) with a predominant forward component and slightly skewed distribution of acute turning angles (Fig. 1c). For both St1 and St2 modes, the time span for positive velocity autocorrelation has been estimated and discussed in the Supplementary Fig. 1 and Supplementary Note 1. St1 remarkably

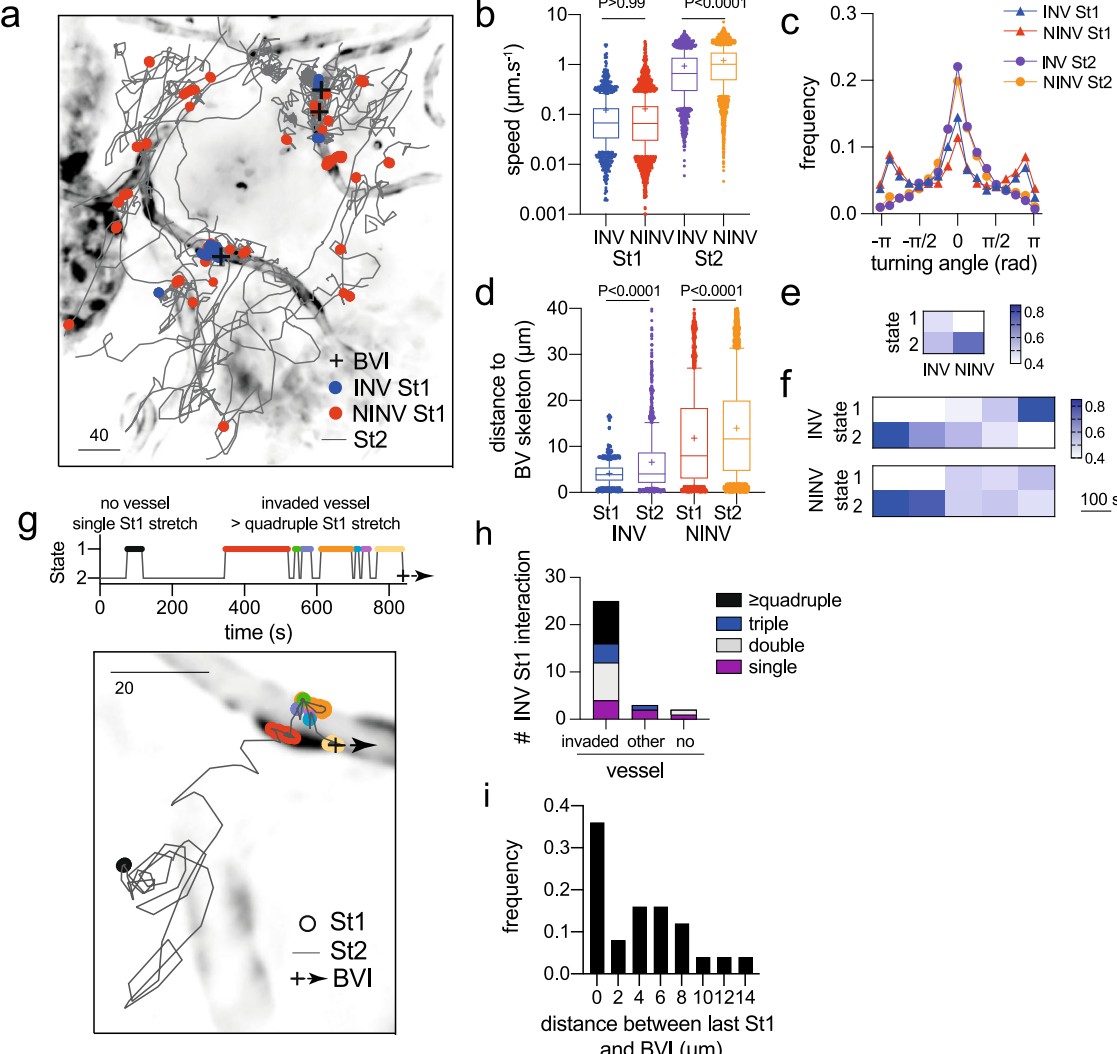

**Fig. 1 | Hidden Markov Model (HMM) analysis of sporozoite motility. a** Typical parasite tracks (scale bar: 40 μm) showing state 1 motility (St1, circles) of blood vessel (BV) invaders (INV, blue) and non-invaders (NINV, red) strikingly co-localizing in the same BV areas. HMM state 2: gray lines, BV invasion sites (BVI): black crosses. **b** Quantification of sporozoite velocity showing low-speed St1 and high-speed St2 motility. **c** Frequency of sporozoite turning angle distribution. **d** Distance to the nearest BV according to HMM states and invasive phenotype. **e** INV/NINV HMM states distribution (INV = 2819; NINV = 11,581; Two-tailed Fisher's exact test $P < 0.0001$). Darker shades of blue indicate increasing frequencies. **f** State frequencies over time for 24 INV/NINV sporozoites. Darker shades of blue indicate increasing frequencies. **g** Example of successive INV St1 interactions. St1: colored circles, St2: gray line, BVI: black cross and dashed arrow. Upper graph: state

alternation over time with colored St1 stretches matching the image below. Image: INV example showing an extravascular St1 stretch (black circles) and multiple St1 stretches (≥quadruple, colored circles) on an invaded vessel. **h** Number of INV displaying St1-stretches close (invaded vessel) or far (other vessel) from BVI or within the extravascular tissue (no vessel). Colors indicate the number of St1 stretches performed by each INV sporozoite. **i** Frequency distribution of the distance between the last INV St1 and BVI. For parasites flushed with the bloodstream after the last St1, the distance was considered zero. **b, d** Multi-comparison using Kruskal–Wallis/ Dunn's test. Lines denote the median, crosses the mean, and whiskers indicate the 10th and 90th percentiles. **b–i** Data from four independent experiments. Source data are provided as a Source Data file.

identified parasite interactions with BV for both INV and NINV (Fig. 1a, d), including in the zone of BV invasion (BVI, Fig. 1a, black crosses). These quantifications are in good agreement with previous observations of avascular high-motility and vascular low-motility migration patterns of sporozoites in the dermis[5,8]. St1 was also frequently identified farther from BV in NINV, revealing additional low motility interactions with host structures in the extravascular tissue (Fig. 1a, d, red symbols).

INV spent 47% of their time in St1 while NINV only 34% (Fisher's exact test, $P \leq 0.0001$; Fig. 1e). When the state frequencies of parasites were quantified over intervals of 100 s during the first 500 s of their tracks, a clear transition from a prevailing St2 to a predominant St1 behavior ending up with an intravasation was observed over time for INV ($n = 24$, Fig. 1f, INV). The initial behavior of NINV was similar to INV, but instead of engaging in a preponderant St1 low-motility mode, NINV returned to a predominant St2 high-motility mode ($n = 24$, Fig. 1f, NINV). Intriguingly in 78% of INV tracks, St1 was distributed in multiple stretches along the invaded vessel, alternating with contiguous sequences of St2 (Fig. 1g, circles, invaded vessel). These discrete stretches of St1 suggest a local scanning behavior, hinting at the presence of dispersed perivascular cues, which may promote motility switch and invasion. Accordingly, observed sequences of multiple St1 interactions mainly occurred along portions of invaded BV (Fig. 1h). In 88% of invasion events, which were scored when sporozoites were flushed with the bloodstream after interacting with a vessel, the last St1 point was less than 10 μm apart from the BVI site, indicating this point probably marks or is very close to the site of intravasation, and that the process of BV entry is strongly associated with St1 motility. In the remaining cases, sporozoites shortly moved inside the vessel (<15 μm) before being carried away in the circulation (Fig. 1i).

Altogether, the HMM analysis shows that the sequence of NINV states is compatible with a predominant high-motility search mode (Fig. 1f, NINV, St2% > St1%, first 200 s), followed by an increase in St1 behavior upon BV encounter or extravascular interaction. After an unsuccessful intravasation, NINV return to a high-motility search mode such as in a trial-and-error approach. The sequence of INV states is compatible with an initial high-motility general search mode similar to NINV behavior (Fig. 1f, INV, St2% > St1%, first 200 s). However, upon BV encounter, INV switch states and perform a low-motility, intermittent, local vessel scan (St1% > St2%), which ultimately results in a BV invasion.

## Sporozoites in high-motility mode display a superdiffusive Lévy walk pattern

Lévy walks, which sometimes are indiscriminately and misleadingly called Lévy flights in the literature (see Supplementary Note 2), are very common in the description of superdiffusive foraging behavior[1,2]. To a great extent[18], for a large enough maximum length L, e.g., usually the largest linear length in the searching region, the essential aspects of Lévy walks can be described by a truncated power-law distribution of the step length $\ell$: $P(\ell) \sim C/\ell^\mu$ for $\ell_{min} \leq \ell \leq L$, with $\ell_{min}$ a lower cutoff and C a proper normalization constant[1]. Decreasing μ values within the interval $1 < \mu < 3$ translates into an increasing degree of superdiffusiveness. The limit $\mu \rightarrow 1$ leads to ballistic behavior consisting essentially of extremely large consecutive concatenated steps, while $\mu \geq 3$ corresponds to typical normal diffusion, characteristic of Brownian-like motion[1] (Supplementary Fig. 2).

Owing to the low fractional volume of the microvasculature in our experimental setup (7.2 ± 0.3%, Supplementary Fig. 3), the relative low rate of BV invasion, and the plasticity of Lévy walks, considered as one of the simplest mechanisms allowing behavioral adjustments as response to the environment[19], we hypothesized that parasites in high-motility mode perform a superdiffusive Lévy walk to optimize the finding of intravasation sites.

In spite of the wealth of studies employing Lévy walk models, their identification in experimental data can be a challenging task when the trajectories are two- or three-dimensional and span less than two orders of magnitude[20]. Differently from species that can travel long distances[1,10], sporozoites perform a cutaneous migration limited in space and time by their biology. Accordingly, parasite speed and, thus, displacement decrease over time as well as BV invasions, which mostly occur within 40 min after *P. berghei* sporozoite inoculation into the skin[5]. To optimize the recording of longer trajectories and improve data analysis, we tracked 74 additional sporozoites in a volume of $664 \times 901 \times 30$ μm$^3$ for 40 min, representing an approximately sixfold increase in the area of observation. To approach the problem of the three-dimensional analysis, we used protocols allowing the separation of a given path in its distinct directions, thus considerably facilitating the characterization of the movement turns[21,22]. It is also known that the relative advantage of optimal Lévy searches decreases in three-dimensional search spaces due to the smaller oversampling of random walk trajectories[23–27]. Of note, in our experimental setup, while searching for a BV, sporozoites move in a nearly two-dimensional path in the upper dermis, with thickness (30 μm) considerably smaller than the tissue surface extension (320–664 × 320–901 μm).

Taking into account all the above considerations, the HMM track analysis was further refined to test our hypothesis (see ref. 28 and "Methods"). For each INV and NINV trajectory, segment stretches presenting high- and low-motility modes as assigned by the HMM state analysis were identified (Fig. 2a). The corresponding state alternations over time are depicted in Fig. 2b for four illustrative examples. Step lengths, defined as the distance traveled between two consecutive direction changes in one dimension, were next computed for each track[22] and combined in four groups to generate larger datasets for statistical analysis (INV St1/low- and St2/high-motility and NINV St1/low- and St2/high-motility). A continuous power-law (PL) distribution was fitted to each group individually or to the combined St1+2 groups for INV and NINV (Fig. 2c–f, black dashed line). To test other potential mechanisms leading to superdiffusion, like fractional Brownian motion[29], we additionally fitted an exponential (EXP) distribution[30,31] to the same groups (Fig. 2c–f, cyan dotted-dashed line). Using Akaike Information Criterion (AIC), we found that PL is comparatively the best distribution for the high-motility St2 mode, with μ INV = 1.6 and μ NINV = 2.6, which are exponents leading to a superdiffusive Lévy-like walk (Fig. 2c, e). Conversely, for both INV and NINV at low-motility St1 mode, step lengths are best fitted by an EXP distribution (Fig. 2d, f, insets). The step length range is very small for the St1 mode, which displays rather short steps, and thus has a different distribution trend compared to the St2 mode (Fig. 2d, f, arrows). Importantly, the optimal $\ell_{min}$ fitting parameter for the St2 mode data is not substantially altered by adding the St1 mode data (Fig. 2c, d for INV and Fig. 2e, f for NINV), implying that longer steps are almost absent for sporozoites in the St1. Consequently, PL is the best fit for both INV (Fig. 2c, d) and NINV (Fig. 2e, f) parasites, which display similar superdiffusive μ exponents (μ < 3) for both St2 alone and combined St1+2 distributions.

Interestingly, the dual motility behavior of sporozoites—alternating low (St1) and high (St2) motility modes and resulting globally in a superdiffusive Lévy-like walk—displays some similarities with other biological processes. For instance, the transport of ribonucleoproteins inside neurons also displays two alternating movement states, a very slow and a ballistic mode of transport. Their combination can also result in a superdiffusive Lévy-like walk by convoluting the time duration of each state[32]. An important future investigation should be to determine which molecular mechanisms could account for the St1 and St2 transition in the sporozoite search strategy. A study of intracellular cargo transport has shown that Brownian-like steps allied to self-organizing directional reinforcement may give rise to Lévy walks with μ <3[33]. Although the angle distribution for sporozoites moving in St2 mode might be an indication of directional reinforcement (Fig. 1c, high

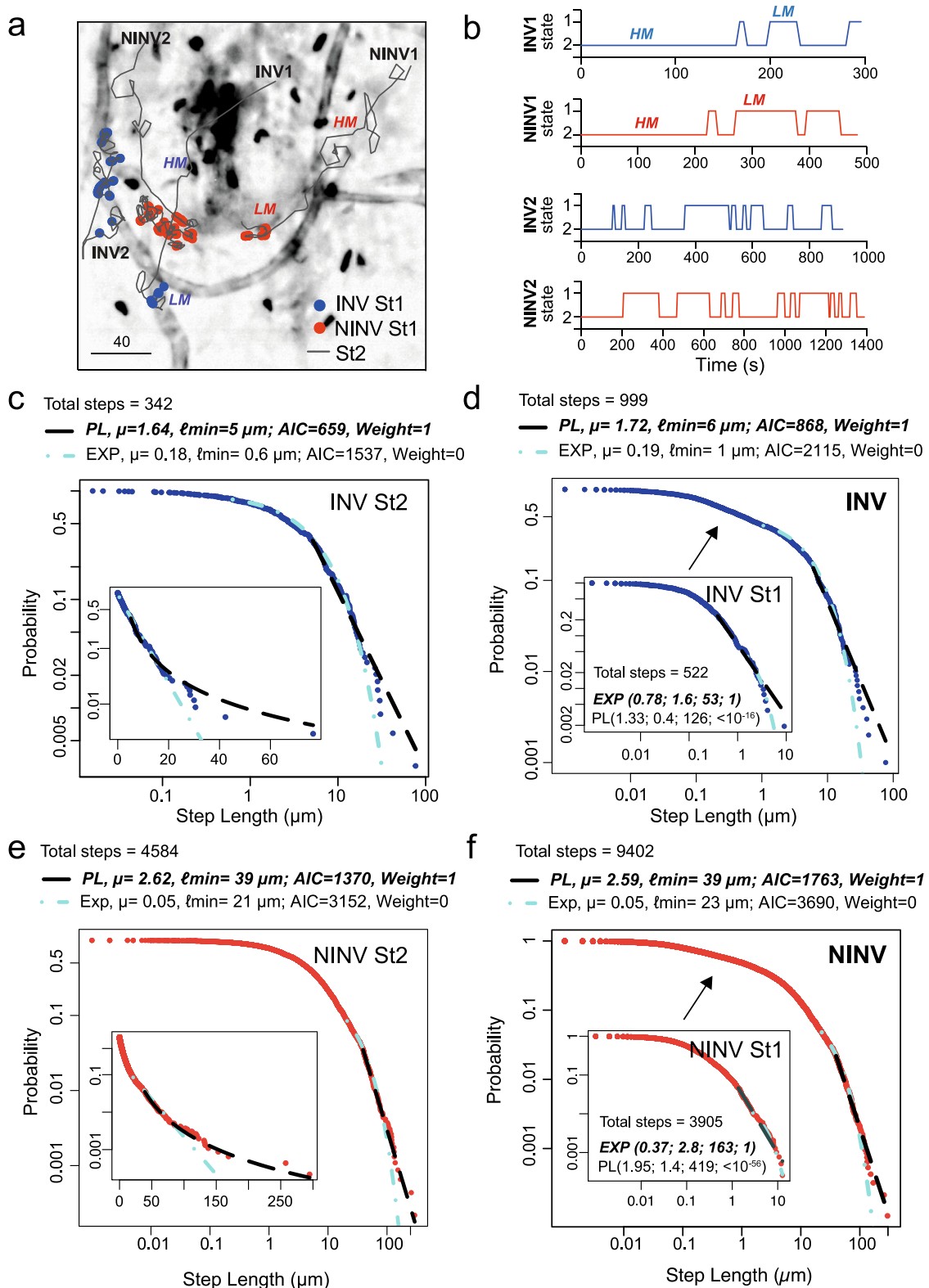

**Fig. 2 | High-motility mode and superdiffusive Lévy walk. a** Representative examples of invader (INV, blue) and non-invader (NINV, red) parasite tracks showing continuous segments of state 2 (St2) high-motility (HM, lines) and state 1 (St1) low-motility (LM, circles). Scale bar, 40 μm. **b** HM and LM state alternation over time for two INV and two NINV exemplary trajectories. **c–f** The best power-law (PL) and exponential (EXP) fits for **c** INV parasites in the St2 mode, **d** INV parasites combining the St1 and St2 modes, **e** NINV parasites in the St2 mode, and **f** NINV parasites combining the St1 and St2 modes. The insets in (**d**) and (**f**) show the fits for St1 mode alone, while insets in (**c**) and (**e**) display log-normal plots of the St2 mode, helping to highlight the long-tail distribution of the experimental data. Circles (INV: blue, NINV: red) show empirical data; dashed (black) and dotted-dashed (cyan) lines, respectively, represent the best PL and EXP fits. The adjusted parameters μ and ℓmin, as well as the total number of steps analyzed, are indicated on each graph. Akaike Information Criterion (AIC) values and weight indicate the best fit highlighted using bold italic letters for each dataset. In the insets in (**d**) and (**f**), numerical values correspond to (μ; ℓmin; AIC value, AIC weight). **c–f** Pooled data from seven independent experiments, yielding 27 INV tracks and 161 NINV tracks. Source data are provided as a Source Data file.

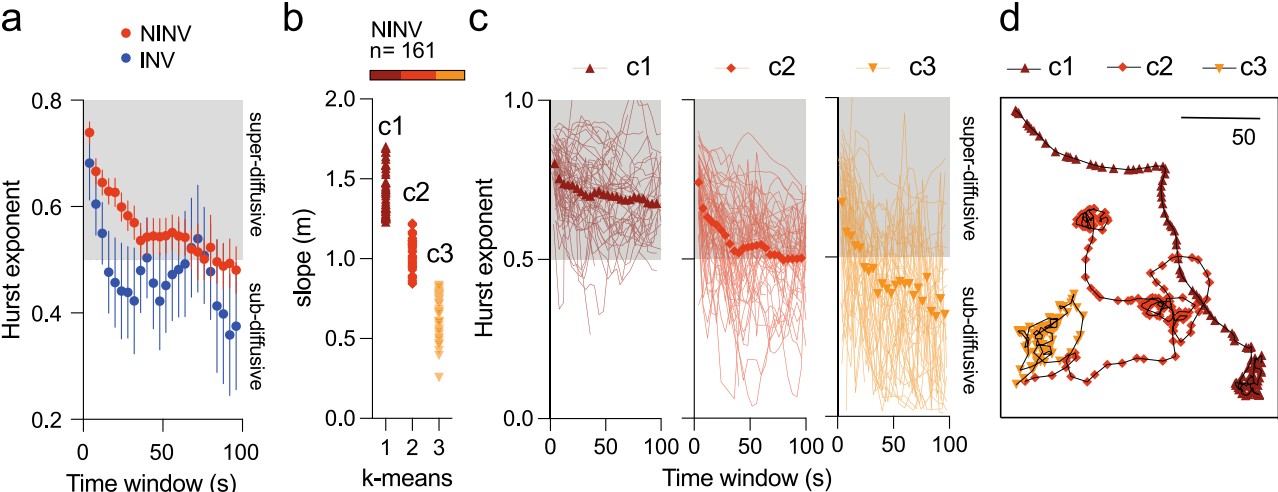

**Fig. 3 | Anomalous diffusive behavior of sporozoites. a** Hurst exponent of blood vessel invaders (INV, blue circles) and non-invaders (NINV, red circles). Symbols represent the average of sporozoites (27 for INV and 161 for NINV) from seven independent experiments ±95% confidence interval. $H > 0.5$, superdiffusion. $H < 0.5$, subdiffusion. **b** k-means clustering of the slope (m) of MSD plotted in log-log scale against the analysis time window for the NINV population. Top graph represents the parasite percentage in each cluster (c1: brown, c2: red, c3: orange).

**c** Hurst exponent time series for sporozoites from clusters c1 (brown), c2 (red) and c3 (orange). Symbols represent the average, and lines represent individual parasites. **d** Representative examples as shown in (**c**) of a superdiffusive, forward-moving parasite (c1, brown), a subdiffusive, meandering, circularly moving parasite (c3, orange), and an intermediate behavior between the two previous phenotypes (c2, red). Bar = 50 µm. Source data are provided as a Source Data file.

frequency of turning angles = 0 rad, characteristic of parasites moving in consecutive steps without direction change), whether this mechanism is involved in the superdiffusive Lévy-like walk of sporozoites still needs to be determined. Finally, from a more technical point of view, recently, machine-learning methods have been successfully used to pinpoint Lévy walks and other anomalous diffusive modes in distinct datasets. They likewise might be helpful to study potential changes of diffusive modes within parasite trajectories[34–36].

## Circular and forward motility are associated with sub- and superdiffusion

To further characterize the overall dispersive behavior of INV and NINV populations, we considered their Hurst exponent ($H$)−a convenient measure of diffusiveness[1,2]. Briefly, for long enough times, the mean square displacement (MSD) scales with time as MSD $\sim t^{2H}$, where $H$ is the Hurst exponent and $t$ is the elapsed time. In turn, MSD is the square of the straight line from the initial departure reference point until the individual position at the time instant $t$, averaged over many realizations. The $H$ is then computed as a function of the temporal span of analysis, i.e., a sliding averaging time window running across the whole duration of the different tracks (Fig. 3, for details, see "Methods"). Trajectories with sub-, normal and superdiffusion are typically characterized by $H < 0.5$, $H = 0.5$, and $H > 0.5$, respectively. In the NINV population (Fig. 3a, red circles, $n = 161$ sporozoites), superdiffusion ($H > 0.5$) is observed at time scales less than 80 s. This could be interpreted as the typical time required for some sporozoites to travel between blood vessels and switch their motility strategy to a less-diffusive type of motion. At larger time scales, the global motile behavior of sporozoites tends to $H \sim 0.5$, typical of normal diffusion and an expected time limit if the underlying Lévy-like processes are spatially truncated[1], which is the case here. Conversely, in the INV population (Fig. 3a, blue circles, $n = 27$ sporozoites), subdiffusion ($H < 0.5$) is predominantly observed at time windows of $t > 16$ s, reflecting the BV-interacting, low-motility mode of INV sporozoites.

It has been observed that prokaryote, microzooplankton and mammalian cells intertwine circular and forward movements to efficiently locomote[37–39]. To better understand the contribution of these two movement patterns to the dispersive behavior of sporozoites, the NINV population was decomposed into three groups based on the

slope of the MSD plotted in log-log scale against the analysis time window using k-means clustering (Fig. 3b, Supplementary Note 3 and Supplementary Fig. 4). Cluster 1 (c1) grouped parasites with a preferential tortuous-forward movement with predominant superdiffusive mean $H$ at all time scales (Fig. 3c, d, c1). Parasites of cluster 3 (c3) were superdiffusive at time scales <20 s and predominantly subdiffusive at time scales >20 s, showing an important circular-meandering component in their trajectories (Fig. 3c, d, c3). Cluster 2 (c2) parasites displayed a mixed behavior of c1 and c3 parasites (Fig. 3c, d, c2).

Altogether, the data reveal a strong superdiffusive component in a long-, medium- and short-time scale range, respectively, for the forward-moving (c1), forward-meandering (c2) and circularly meandering (c3) sporozoite populations. For extra analysis regarding the MSD, see Supplementary Figs. 5, 6, and 7 and the accompanying discussion in Supplementary Note 3. The superdiffusive Lévy-like high-motility mode allows for probing a large portion of the search space, compatible with a global search exploration for BV. On the other hand, subdiffusive behavior is clearly associated with circular motility as observed in INV sporozoite moving around vessels or c3 NINV, suggesting that parasites are performing local exploitation, which allows refining the search to detect targets scattered in a restricted area. Subdiffusive local exploitation thus suggests that the microvasculature is not homogenously permissive to sporozoite intravasation.

## Sporozoites intravasate in hotspots characterized by the presence of a pericyte

In good agreement with the hypothesis that sporozoites actively search for permissive intravasation sites, mapping of BV invasions revealed that ~70% of intravasations occurred within a maximal distance of 20 µm from their closest neighbors, i.e., approximately twice the length of a sporozoite, and sometimes even at the same place. Additionally, most of the invasion sites respecting that distance threshold were distributed in clusters of 2 to 5 entry points circumscribed in circles of less than 25 µm of diameter, thus highlighting the existence of intravasation hotspots along the microvasculature (Fig. 4a, b). Topologically, invasion sites were frequently located in the vicinity of vessel branching points (Fig. 4, c), suggesting that these

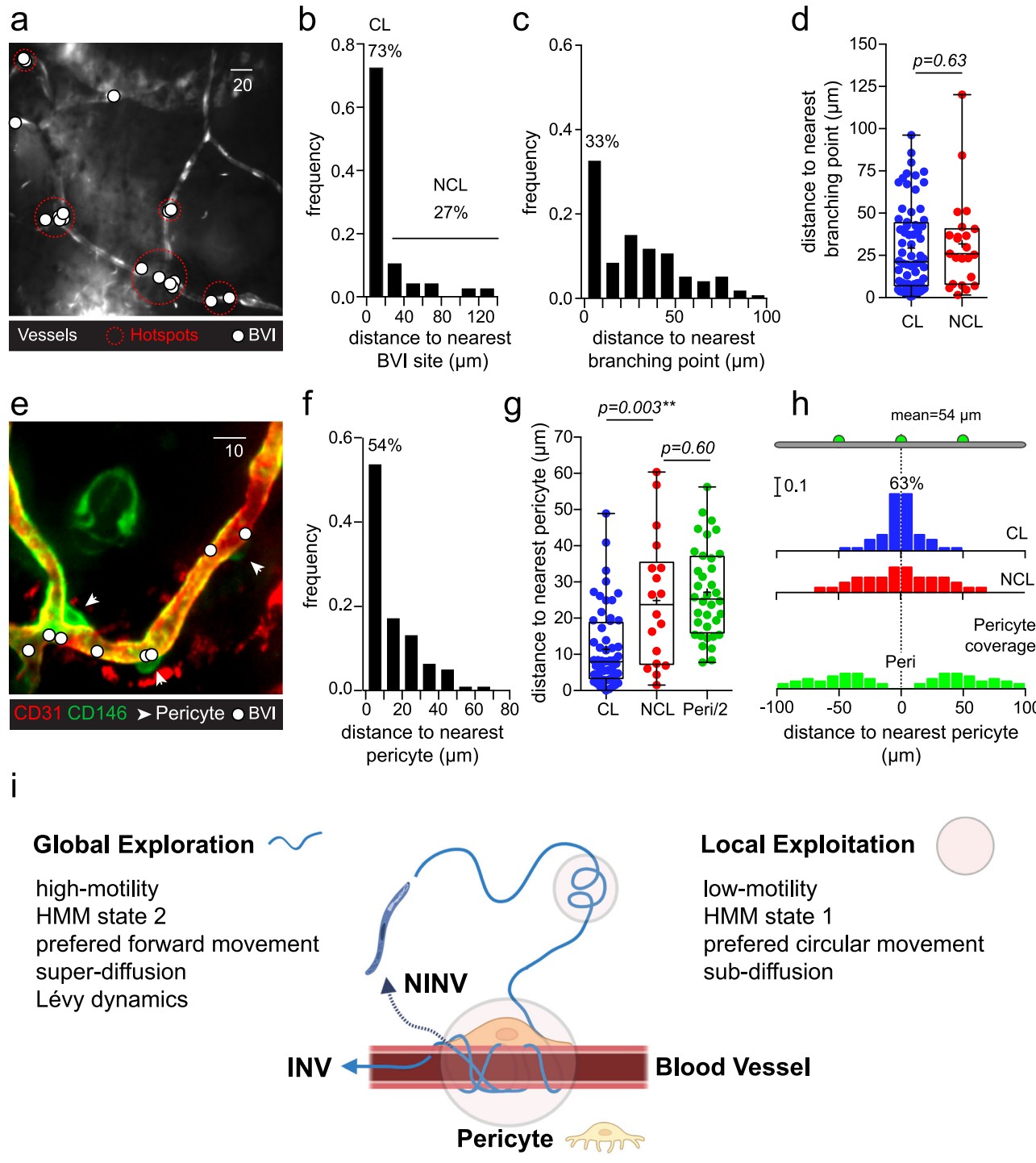

**Fig. 4 | Invasions are clustered at hotspots associated with the presence of pericytes. a** Distribution of blood vessel invasions (BVI, white circles) along the cutaneous microvasculature. Clusters of BVI are indicated by red dotted circles, vessels and sporozoites appear in white. Bar, 20 μm. **b** Frequency distribution of the distance between sites of BVI and their nearest neighbor for clustered (CL) and non-clustered (NCL) invasions. **c** Frequency distribution of the distance between sites of BVI and the nearest vascular branching points. **d** Distance between the sites of BVI and nearest branching points for CL (blue) and NCL (red) invasions. Two-tailed Mann–Whitney test. **e** Distribution of CD146⁺ (green) CD31⁻ (red) pericytes (arrowheads) and clustered BVI (white circles). Bar, 10 μm. **f** Frequency distribution of the distance separating sites of BVI from the nearest pericyte body. **g** Distance between sites of BVI and the nearest pericyte body in CL (blue circles) and NCL (red

circles) populations, or half of the distance between two adjacent pericyte bodies (Peri/2, green circles). Kruskal–Wallis with Dunn's multiple comparison test. **h** Schematic representation of BVI and pericyte distribution. Blue: CL BVI, red: NCL BVI, green: pericyte. **i** Search strategy of sporozoites to intravasate into the dermal microvasculature. HMM: Hidden Markov Model, INV: invader, NINV: non-invader. Created with BioRender.com. **d**, **g** Lines denote the median, crosses the mean, and whiskers the spread of the dataset. Pooled data from five independent experiments yielding 64 BVI events for (**b**) and from 13 independent experiments for (**c**, **d**) and (**f**–**h**) analyzing respectively 91 (**c**, **d**) and 74 (**f**–**h**) BVI. Pericyte coverage (**g**, **h**) was quantified over 10 independent fields containing a total of 47 pericytes. Panels (**a**) and (**e**) are representative of 5 and 13 experiments, respectively. Source data are provided as a Source Data file.

bifurcation points could be associated with preferential invasion sites. However, no significant difference was observed when comparing the distances separating branching points from clustered and non-clustered invasion sites (Fig. 4d, $P = 0.63$). Another element positioned, but not exclusively, at capillary bifurcations may thus define invasion hotspots.

Pericytes are contractile cells that surround endothelial cells in the microvasculature, frequently sitting on endothelial cell junctions[40]. They play an important role in vessel formation, vascular permeability, control of blood flow, and cell transit across vessel walls[41]. Notably, pericytes, whose cell bodies can be visualized using a fluorescently labeled anti-CD146 monoclonal antibody[42] and display a $CD146^+ CD31^-$ phenotype, are located at vascular branching points as well as along blood vessels (Fig. 4e, white arrowheads). To identify a possible association between pericytes and invasion hotspots, the distances separating clustered or non-clustered intravasations sites from the nearest pericyte body were measured, revealing that most invasions occurred close to pericytes (Fig. 4e, f, Supplementary Movie 2); 63% of clustered invasions sites were located within 10 μm of a pericyte body (Fig. 4g, h), whereas non-clustered intravasations occurred on average at a distance similar to the mean half distance between two pericytes (Peri/2, ~25 μm, $P = 0.60$, Fig. 4g). These results indicate that pericytes are associated with intravasation hotspots ($P = 0.003^{**}$), while non-clustered invasions appear randomly distributed in the vasculature (Fig. 4g, h). Whether pericytes are responsible for directing sporozoites to these hotspots remains to be determined. Interestingly, based on in vitro binding experiments, hepatic pericytes, known as stellate cells, have been proposed to play a role in the recognition and arrest of blood-circulating sporozoites in the liver sinusoids[43]. Molecular interactions between sporozoite surface adhesins and pericyte molecules have also been identified[43,44], hinting at the specific binding of sporozoites to pericyte-secreted extracellular matrix or surface molecules exposed either to the blood circulation via the fenestrated liver sinusoids or on the abluminal side of cutaneous BV. These specific interactions could be involved in the switch of sporozoite motility behavior at the endothelial barrier interface allowing the successful crossing of these cellular barriers in the skin and liver.

## Global and local intermittent search strategy to find intravasation hotspots

To summarize, the sporozoite BV search and invasion dynamics can be described as the following (Fig. 4i). Once in the skin, sporozoites start moving in a superdiffusive, Lévy-like, high-motility mode with a preferential forward component. This behavior is consistent with a global exploratory search, seeking efficiently to encounter a BV in the host dermis. Upon BV encounter, sporozoites switch to a subdiffusive low-motility mode with an important circular component. This change is compatible with local vessel exploitation, which allows for more careful scanning of a limited region of BV while probing for a permissive entry site in the microvasculature. These privileged sites of intravasation are associated with the presence of pericytes, which might directly or indirectly guide sporozoites to these hotspots of invasion or simply highlight weaker points for passage across the endothelial barrier. If, after some time, intravasation is not achieved, sporozoites tend to shift back to the superdiffusive mode. The alternation between a global exploration and a local exploitation strategy is a pattern observed in many other biological search processes. Bumblebees use it to learn about proper foraging routes, optimizing energetic costs[45]. In fragmented landscapes, this alternation may also be a mandatory survival behavior for small predators, being themselves prey to other larger predators[11,12]. At a subcellular level, switching between 3D exploratory diffusion and 1D Brownian scanning exploitation, known as enhanced diffusion, can improve the efficiency of DNA binding proteins to find specific sequences in DNA strands[46]. Since organism fitness depends on search efficiency[1,3,9,10], optimizing

its duration and energetic expenditure becomes essential. When the search landscape demands both exploration and exploitation dynamics[11,12,38,46,47], finding the best trade-off is, in fact, crucial for encounter rate optimization.

Hence, sporozoites, grounded on the capacity to move forward and circularly, seem to have evolved a search strategy where they alternate between a global superdiffusive, Lévy-like high-motility mode to find blood vessels and a subdiffusive low-motility mode to find hotspots of invasion associated with pericytes. This super- and subdiffusive search strategy is potentially directed by geometrical constraints linked to specific molecular interactions with host cues. Intermittent search strategies—extensively reported at macroscopic and molecular levels—are now unveiled for a unicellular pathogen.

# Methods

## Ethical statement
Work on animals was performed in compliance with French and European regulations on the care and protection of laboratory animals (EC Directive 2010/63, French Law 2013-118, February 6th, 2013). All experiments were approved by the Ethics Committee #89 (CETEA, Institut Pasteur) and registered under reference 01324.02, 32422 and 32989 by the French Ministry of Higher Education, Research and Innovation.

## Parasites, mosquitoes, and mice
All experiments were performed using cloned lines of *P. berghei* ANKA expressing either the GFP under the control of the *hsp70* promoter[48] or the RedStar fluorescent protein under the control of the *eef1-a* promoter (line L733)[49].

Female *Anopheles stephensi* mosquitoes (SDA 500 strain) were reared in the Center for the Production and Infection of Anopheles at the Institut Pasteur and were allowed to feed on infected mice 3–4 days after emergence. Mosquitoes received a non-infectious blood meal 7 days after infection and were kept in a climatic chamber at 21 °C, 80% humidity under a 12-h dark/light cycle with sucrose ad libitum as previously described[14]. Sporozoites were isolated from salivary glands 21–28 days after the infectious blood meal and kept on ice in PBS until inoculated into mice.

Specific pathogen-free RjOrl:SWISS and C57BL/6JRj mice were purchased from Janvier Labs (Le Genest Saint Isle, France), and *flk1*-GFP transgenic mice[50] on a C57BL/6 background were bred under specific pathogen-free conditions at the Institut Pasteur. Female RjOrl:SWISS mice aged 4–6 weeks were used to provide infectious and non-infectious blood meals to mosquitoes. All experiments were performed on C57BL/6JRj or *flk1*-GFP female mice aged 5–10 weeks using groups of 4–13 mice, depending on the type of analysis performed. Animals were housed in individually ventilated cages in the animal facilities of the Institut Pasteur, all accredited by the French Ministry of Agriculture for performing experiments on live rodents. Mice were kept under standard conditions (10/14-h dark/light cycle, temperature $20 \pm 2$ °C, humidity $50 \pm 10\%$) with irradiated rodent feed and autoclaved water ad libitum. Mouse sex was not considered a relevant variable for this study since intradermal sporozoite challenge yields comparable levels of infection in male and female mice.

## Intravital spinning-disk confocal microscopy
Intravital imaging was performed using a spinning-disk confocal system (UltraView ERS, Perkin Elmer) composed of four Diode Pumped Solid State Lasers (excitation wavelengths: 405 nm, 488 nm, 561 nm and 640 nm), a Yokogawa Confocal Scanner Unit CSU22, a Z-axis piezoelectric actuator and a Hamamatsu EMCCD C-9100 camera mounted on an Axiovert 200 microscope (Zeiss) controlled by Volocity (Perkin Elmer). To identify blood vessels and $CD31^- CD146^+$ pericytes, anti-mouse CD31 (clone MEC 13.3 or 390, Biolegend) and anti-mouse CD146 (clone ME-9F1, Biolegend) antibodies coupled to Alexa Fluor®

488, Phycoerythrin or Alexa Fluor® 647 were injected either intravenously immediately prior to imaging (CD31, 15–20 µg) or intradermally 4–6 h before the acquisition (CD31, CD146, 2–4 µg). For intradermal staining, mice were lightly anesthetized with a mixture of ketamine (40–50 mg/kg body weight, Imalgene 1000, Merial) and xylazine (4–5 mg/kg body weight, Rompun 2%, Bayer) and antibodies were inoculated at multiple sites within the dorsal sheets of the ear pinnae using a 10-µL NanoFil microsyringe mounted with a 35G beveled needle (World Precision Instruments). For intravital imaging, mice were anesthetized either with isoflurane (3% for induction, 1% for maintenance) or with a mixture of ketamine (62.5–125 mg/kg body weight) and xylazine (6.25–12.5 mg/kg body weight). Ears were then gently epilated with a piece of tape, and 0.2 µL of a suspension of salivary gland sporozoites were injected intradermally under a stereomicroscope using the previously described NanoFil syringe. Inoculation sites were then immediately imaged. During acquisition, animals were kept warm with a heating blanket (Harvard Apparatus), and their anesthesia status was frequently monitored[14]. Z-stacks covering 30 µm depth were acquired using a Plan-Neofluar 10×/0.3 objective (Zeiss), an LCI Plan-Neofluar 25×/0.8 Imm Korr DIC objective (Zeiss), an EC Plan-Neofluar 40×/1.3 oil objective (Zeiss) or a Plan-Apochromat 63×/1.4 oil objective (Zeiss). Track analysis was performed on projection movies of six Z-stacks acquired with a frame interval of 4 s.

## Image analysis and quantification

Image files were processed and quantified using Fiji[51] (version 2.0.0 to 2.9.0). Parasite tracks were manually defined using the mTrackJ plugin (version 1.5.1). Blood vessel invasion was scored when, after interacting with a vessel wall, a sporozoite could be seen flushed in the bloodstream. To characterize the distribution of invasion sites, the vascular segments irrigating the different fields of view were delineated manually by taking advantage of the fluorescence signal of transgenic or CD31-labeled endothelial cells, and a 1-pixel wide skeleton of the vascular tree was extracted from these outlines. All positions in the corresponding field associated with an event of blood vessel invasion were then projected on the skeleton according to the smallest distance between the invasion site and the skeleton using a custom-made macro. Finally, the lengths of the skeleton segments separating each site of entry from its closest neighbor were measured using the Lines8 plugin developed by Gabriel Landini (version 2.13). To characterize the topology of invasion sites, the positions of the branching points were assigned directly to the extracted skeletons. CD31⁻CD146⁺ pericyte cell bodies, on the other hand, were delineated manually, and their position was defined by projecting on the skeleton the centroids of the defined structures. Distances between the sites of blood vessel invasion and the closest branching points or pericytes were quantified using the positions of the corresponding projections and were measured along the path followed by the skeleton using the Lines8 plugin developed by Gabriel Landini (version 2.13).

## Identification of high- and low-motility states

Each individual $x$ and $y$ time series were independently analyzed with the moveHMM package (version 1.0.1) of the R software[17] with the goal of identifying two different motility states. For each state, the initial reference parameters of mean step length, standard deviation, mean turning angle and angle concentration, drawn from a von Mises distribution of turning angles, are taken as:

  i. Mean step length: 0.1 (St1), 0.5 (St2);
  ii. Standard deviation of step length: 0.1 (St1), 0.5 (St2);
  iii. Mean turning angle: pi (St1), 0 (St2);
  iv. Angle concentration: 1 (St1), 1 (St2).

The initial parameters are not critical in obtaining the results. In fact, they are progressively fitted by the HMM algorithm until reaching a convergence from the maximum likelihood function. We identified

the HMM state 1 (St1) as generally displaying low mean and standard deviation of step lengths, while the HMM state 2 (St2) presented values up to fivefold larger for these quantities. With the states identified, each time instant of the $x$ and $y$ time series was labeled with the motility state performed by the sporozoite at that time instant.

## Power-law and exponential fits

Initially, each trajectory was divided into segments for analysis. For a given trajectory, the coordinate points $x$ and $y$ were independently analyzed to identify time instants in which changes in movement direction occurred. The displacement between two consecutive direction-changing time instants was defined as a step. If all time instants between the direction-changing time instants were labeled with the same HMM label (St1 or St2), then the step was also labeled as St1 or St2; if not, then the step was labeled as mixed motility mode. Mixed motility mode steps were not included in the power-law fit. St1 and St2 steps were then further categorized into four groups (invasive St1, non-invasive St1, invasive St2 and non-invasive St2). Then, the poweRlaw package[28] was used to estimate the best-fitting power-law and exponential parameters (exponent and minimum value) for the step length distribution of each group.

The Akaike Information Criterion (AIC) weight was calculated to compare power-law and exponential fits[52]. For each case, the negative log-likelihood quantifier

$$NLL_i = -\log[L_i] \tag{1}$$

was computed also using the poweRlaw package. Here $L_i$ is the likelihood for the distribution $i$ (where $i =$ power-law or exponential). In this way, an AIC for each case is obtained as

$$AIC_i = 2k + NLL_i \tag{2}$$

$k = 2$ is the total number of compared models. From the values of AIC, the AIC weight $w_i$ was calculated through

$$w_i = \frac{\exp[-\Delta_i]/2}{\sum_{j=1}^{2}\exp[-\Delta_j/2]} \tag{3}$$

With

$$\Delta_i = AIC_i - \min(AIC) \tag{4}$$

for min($AIC$) the minimum AIC over both cases. Therefore, $w_i$ can be interpreted as the probability that the distribution $i$ is the best for describing the data.

## Hurst exponent

The Hurst exponent for a set of $N$ sporozoites was calculated using Python by first calculating the Hurst exponent $H_i$ of each individual $i$, and then averaging the $H_i$'s over $i = 1, \ldots, N$ to obtain a collective characteristic exponent. Let $(x_i(t), y_i(t))$ be the time series representing the trajectory of the $i$-th sporozoite in a total number of $L$ steps and recorded at each $t_{step}$ time interval, then the Mean Squared Displacement (MSD) as a function of time window $\tau$, for $\tau = n_{step} t_{step}$ with $n_{step} = 1, 2, \ldots, \frac{L}{4}$, is obtained as

$$MSD_i(\tau) = \frac{1}{L - n_{step}} \sum_{t = t_{step}}^{t_{step}(L - n_{step})} \left[(x_i(t) - x_i(t + \tau))^2 + (y_i(t) - y_i(t + \tau))^2\right] \tag{5}$$

From the $MSD_i(\tau)$, the Hurst exponent is calculated for each time window $\tau$ as half of the linear coefficient of two consecutive MSD in

log-log scale:

$$H_i(\tau) = \frac{1}{2}\left(\frac{\log MSD_i(\tau + t_{step}) - \log MSD_i(\tau)}{\log(\tau + t_{step}) - \log(\tau)}\right) \qquad (6)$$

From the $H_i(\tau)$, the average Hurst exponent for all $N$ individual was calculated as:

$$H(\tau) = \frac{1}{N}\sum_{i=1}^{N} H_i(\tau) \qquad (7)$$

which forms the set of individual Hurst exponents to be averaged.

## k-means

For the analysis of clustering of the MSD coefficients, we performed a linear fit of the individual MSD versus time window in log-log scale:

$$\log(MSD_i(\tau)) \propto m\log(\tau) \qquad (8)$$

where $m$ is the slope of the MSD shown in Fig. 3b. Next, these values were clustered by the k-means algorithm in Julia[53]. The number of clusters was set to three.

## Statistical analysis and data representation

Statistical analysis was performed using the software GraphPad Prism 9.5.1. Datasets were tested either with a Kruskall–Wallis test followed by Dunn's multiple comparison, a two-tailed Fisher's exact test or two-tailed Mann–Whitney $U$ test with a confidence level of 95%. Graphs were generated with the softwares GraphPad Prism (version 9.5.1) and R (version 4.2.2). Illustrative simulations were obtained with the R adehabitatHR package (version 0.4.20), and cartoons were created with BioRender.com.

## Reporting summary

Further information on research design is available in the Nature Portfolio Reporting Summary linked to this article.

## Data availability

Raw parasite track data generated in this study have been deposited in a public GitHub repository accessible at https://doi.org/10.5281/zenodo.7858654[54]. The processed data resulting from the analysis of intravital movies and parasite tracks are provided with this paper in the Source Data file. Source data are provided with this paper.

## Code availability

All the codes used for the analysis of parasite motility are available online in a public GitHub repository under https://doi.org/10.5281/zenodo.7858654[54].

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

## Acknowledgements

We would like to thank the team of the Center for Production and Infection of Anopheles (CEPIA, C2RA, Institut Pasteur) for providing mosquitoes, the staff of the Central Animal Facility (C2RA, Institut Pasteur) for animal care and breeding, the staff of the Photonic BioImaging platform (UTechS PBI, C2RT, Institut Pasteur) for providing training and access to spinning-disk microscopes. We are grateful to Joana Tavares for sharing reagents and expertise in intravital imaging, Jean-Yves Tinevez for insights on image analysis automation, Dr Gandhi Viswanathan for extensive discussions and critical reading of the manuscript and Dr Robert Ménard for continuous support. This work was supported by funds from the Institut Pasteur, the Agence National de la Recherche (ANR, French National Research Agency)/Deutsche Forschungsgemeinschaft (DFG, German Research Foundation)—project number SporoSTOP ANR-19-CE15-0027, and the French Government's Investissement d'Avenir program, Laboratoire d'Excellence Integrative Biology of Emerging Infectious Diseases—project number ANR-10-LABX-62-IBEID (R.A.), Coordination for the Improvement of Higher Education Personnel (CAPES) – Program for Institutional Internationalization (PRINT) Systems Biology and Omics Sciences Applied to Biosciences and Health grant 88887.311835/2018-00 (M.G.E.L. and R.A.), Conselho National de Desenvolvimento Cientifico e Tecnologico (CNPq) grants 304532/2019-3 (M.G.E.L.) and 305062/2017-4 (E.P.R.), and Fundação de Amparo à Ciência e Tecnologia do Estado de Pernambuco (FACEPE) grant APQ-0602-1.05/14 (E.P.R.).

## Author contributions

P.F. performed intravital imaging experiments. P.F., H.Z. and R.A. extracted sporozoite trajectories from the videos. M.E.W., R.M.T., J.G.P., Y.B.M., H.Z., E.P.R. and M.G.E.L. developed and implemented the methodology and codes to analyze sporozoite tracks. P.F. performed the topological analysis of blood vessel invasion site distribution. P.F. and M.E.W. participated in manuscript writing. M.G.E.L. and R.A. supervised overall data analysis and wrote the manuscript. P.F., M.E.W., R.M.T., J.G.P., Y.B.M., E.P.R., M.G.E.L. and R.A. all contributed to manuscript revision and editing.

## Competing interests

The authors declare no competing interests.
