## [Peer Review File · Nature Communications]

Reviewer comments, first round -

Reviewer #1 (Remarks to the Author):

The authors report a highly interesting study on how malaria parasites enter the human body as Plasmodium sporozoites. They demonstrate that there exist two fundamentally different dynamic patterns, namely, superdiffusive search phases and subdiffusive local exploration. While I have substantial criticism on some of the data interpretation, I am convinced that the paper contains sufficient novel and important information to ultimately qualify for publication in Nature Communication. While I cannot comment much on the data and the biological background, I strongly believe that the general observations in this work are very exciting.

(1) On a more formal level, the authors talk a lot about Levy-like and Levy walk transport, but they do not properly define these processes in the text. They should include this information, along with a brief discussion of which precise process they are talking about. Levy flights are Markovian processes with diverging second moment. Levy walks introduce a finite propagation speed and have converging second moments. See J Stat Phys 27, 499 (1982); Phys Rev A 35, 3081 (1987) for discussions.

(2) It is exactly the claim of Levy walks/flights that I feel most at odds with. The evidence claimed from figure 2e is in my view a massive overstatement. Even in the old days of fractals it was always clear that power-laws with less than a decade of scaling are no power-laws. Here we have 1/2 decade of proper scaling, if at all. Also in the preceding two panels c and d I would contest that the shown power-law fit makes much sense. Instead, the curves to me look much more like an exponential function or related function decaying quicker than a power-law.

Having said this, I would ask the authors to study an alternative scenario that could indeed be relevant here. Namely, in Phys Rev Res 4, 033055 (2022) it was recently observed that searching birds, instead of the long-standing dogma that they perform Levy-type search processes, follow a persistent superdiffusive motion compatible with fractional Brownian motion with Hurst index larger than 1/2. To see this the authors should check how good or bad an exponential fit is to their data in figures 2c-e. Moreover they should check the velocity autocovariance function of the observed motion. For the superdiffusive parts relevant for figure 2e these might be positively correlated, and these correlations should fall off slowly, compare the analysis in the above paper. Given that there is active motion involved in the system such a scenario is not unlikely and could provide a similarly reasonable explanation towards optimised search.

For the subdiffusive parts reported by the authors I would also ask for an analysis of the velocity autocovariance. This might show compatibility with

a viscoelastic behaviour. Alternatively, maybe trapping is relevant, i.e., more of a subdiffusive continuous time random walk scenario? Please show some more details on this part of the motion, as well.

(3) For the reported anomalous diffusion exponents I ask the authors to also show some examples for the mean squared displacements. Otherwise it is quite difficult to assess the relevance of the stated exponents. In particular it would be nice to show both ensemble and time averaged MSDs and discuss their statistics.

(4) I did not understand the statement at the beginning of the paragraph starting in line 52. Does it mean that parts of individual trajectories were discarded? If yes, this could alter the correlations of the associated time series. Please explain and rephrase.

(5) There are many statements about how Levy search can improve the search efficiency in literature. However, this is quite dimension-dependent. In one and two dimensions, there is a clear advantage, however, in three dimensions this is no longer significant, due to the much smaller oversampling of random walk processes in higher dimensions (Phys Rev Lett 124, 080601 (2020)). In one dimension and effectively one dimension (see, e.g., Phys Rev Lett 95, 260603 (2005)) the best-case corresponds to a Levy index of 1, i.e., $\mu=2$ in the present notation. However, small disturbances can significantly reduce this advantage, as analysed in Proc Natl Acad Sci 111, 2931 (2014). These points should be mentioned in the discussion.

(6) There exist examples for Levy walks in cellular systems involving molecular motor motion. These should be included in the discussion: Nature Comm 9, 344 (2018); Nature Mater 14, 589 (2015).

(7) Change point analysis was shown to work nicely in machine learning approaches. The latter have also become very successful in identifying Levy walk motion. Maybe they could be used to analyse the current data. While I do not insist that the authors study this in the framework of the current manuscript, the potential should be mentioned in the discussion, compare Nature Comm 12, 6253 (2021); J Phys A 55, 244005 (2022); Nature Comm 13, 6717 (2022).

(8) In figure 1, could you replace the green crosses by some darker orange? For readers with colour vision problems the green crosses are virtually invisible.

Reviewer #2 (Remarks to the Author):

The new findings of this paper are interesting and an important contribution to the field. Specifically they show that sporozoites, once close to blood vessels, perform a local exploration to detect blood vessel entry points that are scattered within a restricted area characterized by the presence of

pericytes and blood vessel junctions. These data, which appear to be well-supported, bring us a little closer to understanding how sporozoites search for and recognize blood vessel entry points. However, there are two major issues concerning their findings around the two types of motility exhibited by parasites that need to be addressed:

1) Blood vessels in the dermis, which is the tissue where the majority of mosquito probing occurs, are not rare. Indeed capillary density in the dermis is between 50 to 70 capillaries per mm². Given that sporozoites are 10 microns in length, this means that there is a blood vessel within a few sporozoite lengths in any given part of the dermis. Thus, a Levy-walk does not make sense as blood vessels are abundant. What is not abundant are the entry points and I wonder if they are confusing the Levy-like searching that occurs once a parasite has encountered a blood vessel with a Levy-walk that is frequently used by some organisms to find a rare item. The references they use to support that blood vessels are rare in the dermis are not focused on blood vessel density in the dermis and any image of labeled blood vessels in the skin can immediately dispel the notion that they are rare.

2) A previous study that they cite, (Hopp et al., Longitudinal analysis of Plasmodium sporozoite motility in the dermis reveals component of blood vessel recognition. eLife 2015), uses quantitative motility analysis to demonstrate what had been observed in one of this group's previous papers (Amino et al., Quantitative imaging of Plasmodium transmission from mosquito to mammal. Nat Med 2006)^[SEP], namely that sporozoites display two types of motility in the skin: A high speed, more linear motility that maximizes tissue exploration and a low-speed more constrained motility once they are in the vicinity of blood vessels which is likely related to blood vessel recognition. Since it is important that results are reproducible, it is not a problem that this group, using different methodologies, demonstrate the same thing. However, it needs to be clearly stated that these two types of motility, with one being associated with blood vessels, has been previously shown.

Reviewer #3 (Remarks to the Author):

The inoculation of sporozoites into the human host is an absolutely critical stage of the parasite lifecycle, both as the most significant bottle neck in the mammalian host and a vital target of current malaria vaccines. The fleeting nature of this step means that we still understand little of the parasite dynamics/behaviours underpinning these early steps. This work applies intravital imaging approaches with innovative statistical models to understand sporozoite search behaviour. This adds significant insight into how sporozoites are able to both find blood vessels and what conditions are required for intravasation. I think this would be of significant interest to the broad audience of Nature Comms, although would benefit from some greater explanatory detail to ensure it is as accessible to the broad scientific community as possible.

The manuscript is well written with clear concise intro and discussion sections, but I found it a bit hard to follow through some of the results sections. Whilst this may be my shortcoming, given the interdisciplinary nature of work a few sections could do with some additional more basic explanation to make sure it is as accessible as possible and ensure the exciting key findings hit home. I will give a few examples below, but worth authors going through the manuscript themselves with an eye for simplifying technical language (without sacrificing technical content). I would also be keen to see slightly expanded explanation of how detection of a BV affects followed.

Major points

Line 85 – The sentence here indicates that after a failed St1 stretch the sporozoite reverts back to search mode. This didn't make sense to me, as surely once a vessel is found it makes sense to just try somewhere further along the same vessel. That does seem to be what happens for example Fig 1G. Does initial contact then lead to confinement of the search pattern in proximity to the BV?

Fig 1A Similarly there looks to be an interesting ST1 hotspot just down and left from the centre site in fig 1. This looks like it could be on a BV but it looks slightly larger and less defined. Is this also a vessel? Is it different in type to the others (given its twice diameter). This cluster doesn't have an entry point. Are there other such examples?

Fig 4 was there any analysis done to look at the NINV parasites. It would be interesting to establish whether they just failed to find the right (pericyte proximal) place to invade, or whether they were intrinsically less fit for invasion etc. Instead of using invasion site, you could use the ST1 motility mode as a marker to examine whether ST1 mode in either NINV or INV correlates with pericyte position. This could be informative and presumably existing experiments could be reanalysed in this way?

Technical language – I found a few sections hard to follow, particularly sections like lines 94-99 Are quite hard to follow. Phrases like ballistic behaviour are no doubt right but not super obvious in meaning to me. A lay explanation of a levy walk would be useful too (I googled it).

General – Whilst it makes sense to not change formatting between submissions, I think the final version should be divided up into results subheadings. This would help clarify the messages of each section. I also don't see any movie attachments in the version I have. I think adding some of the example images as supplementary video files would really help to visualise the distinct patterns (perhaps with some annotation – for example video for fig 1g with marked phases would be great).

Minor points

L142 – inferior could be replaced with less than?

Fig 3 D colour of C3 legend key is pink instead of orange.

Fig 3 legend "Symbols represent the average of 4 independent experiments" could you define what is meant by independent experiments here or in methods (i.e. is it different mice, different mosquito infections or different videos etc etc.)

All the best,
Rob Moon

[redacted]

Dear Reviewers,

We would like to thank you for the time and attention dedicated to review our manuscript as well as for the suggestions given, aimed to improve the interpretation and discussion of our results.

We shall highlight that to address the raised points, we: (a) performed three additional independent experiments to collect longer tracks to ameliorate the data fit; (b) compared power-law and exponential fit using Akaike Information Criterion, (c) removed the velocity filter and reanalyzed all data related to HMM states. Importantly, this has not introduced any relevant modifications in the results, and associated interpretation; finally (d) added seven novel supplementary figures with associated texts and references (+19 and +9 references in the main text and SI, respectively) and 2 new supplementary movies. Also, more technical discussion, not fundamental to the main text, but pertinent for a broader overview on our theoretical framework was introduced to the SI.

Our answers are marked in blue and the major modifications in the manuscript are marked in red.

REVIEWER COMMENTS

Reviewer #1 (Remarks to the Author):

The authors report a highly interesting study on how malaria parasites enter the human body as *Plasmodium* sporozoites. They demonstrate that there exist two fundamentally different dynamic patterns, namely, superdiffusive search phases and subdiffusive local exploration. While I have substantial criticism on some of the data interpretation, I am convinced that the paper contains sufficient novel and important information to ultimately qualify for publication in *Nature Communication*. While I cannot comment much on the data and the biological background, I strongly believe that the general observations in this work are very exciting.

We would like to thank the Referee 1 for the positive comments about our work. All the criticisms raised are pertinent and below we shall clarify the points mentioned by the

Referee 1. We thank again the Referee 1 for bringing them up, which allowed a substantial amelioration of the manuscript.

(1) On a more formal level, the authors talk a lot about Levy-like and Levy walk transport, but they do not properly define these processes in the text. They should include this information, along with a brief discussion of which precise process they are talking about. Levy flights are Markovian processes with diverging second moment. Levy walks introduce a finite propagation speed and have converging second moments. See J Stat Phys 27, 499 (1982); Phys Rev A 35, 3081 (1987) for discussions.

R1A1: The Referee 1 is correct. Of course, a physicist/mathematician would be quite aware about the importance of these distinctions, also knowing the essential differences. But perhaps, a broader readership as typical in Nature Communications, not. In the paper indeed, we consider a Lévy walk, not a Lévy flight, since as the Referee 1 has noticed, time spent in a step is explicitly considered, as it should be for a spatial-temporal biological process.

Now, a few words have been added to the main text to point out the mentioned distinction (new line 97).

“Lévy walks, which sometimes are indiscriminately and misleadingly called Lévy flights in the literature (see Supplementary Information), are very common in the description of such super-diffusive foraging behavior ^{1,2}.”

Then, a brief overview about Lévy walks is addressed in the main text (very similar to the original one, so not copied here, but with few pertinent modifications).

Further, a referenced definition of Lévy walks and flights is presented in more details in the new **Supplementary Information**, including their common and distinctive features.

The new supplementary text now reads:

Lévy walks and Lévy flights

Lévy walks and flights have been successfully applied to model a wide variety of phenomena displaying super-diffusive dynamics in many fields ^{1,2}. Sometimes these jargons are misleadingly used in the literature as meaning the same kind of process. In fact, although similar in some points, Lévy walks and flights differ in a number of important features. Lévy random walkers and fliers present step lengths drawn from the family of Lévy α -stable distributions ^{1,2}, with the Lévy index in the range $\alpha \in (0, 2]$ (the limit case $\alpha = 2$ corresponds to the Gaussian distribution).

While Lévy flights are Markovian processes with jump duration (generally instantaneous) independent of the length and diverging second moment for $\alpha \in (0, 2)$, steps in Lévy walks are taken with finite (generally constant) speed, thus generating spatio-temporal correlations leading to non-Markovian temporal evolution and converging second moment ^{3,4,5,6,7}. As closed mathematical expressions for Lévy α -stable distributions in terms of simple

functions are known only for a few cases ^{1,2} (e.g., the non-skewed Cauchy distribution with $\alpha = 1$), in many instances power-law distributions of step lengths ℓ in the form $P(\ell) = C/\ell^\mu$, where $\ell \geq \ell_{min}$ and $\mu = \alpha + 1$ in the range $1 < \mu < 3$ (with ℓ_{min} a lower cutoff length and C a proper normalization constant ¹), have been applied to describe super-diffusive dynamics, since $P(\ell)$ corresponds to the asymptotic large- ℓ limit of Lévy distributions with $0 < \alpha < 2$ ^{1,2}. We note that decreasing μ values within the interval $1 < \mu < 3$ translate into an increasing degree of super-diffusiveness, with the limit $\mu \rightarrow 1$ (i.e., $\alpha \rightarrow 0$) leading to ballistic behavior consisting essentially of extremely large steps ¹. The case $\mu = 3$ ($\alpha = 2$) corresponds ¹ to normal diffusion typical of Brownian-like motion (power-law distributions with exponent $\mu > 3$ also display normal dynamics due to the central limit theorem).

SI References

1. Viswanathan GM, da Luz MGE, Raposo EP, Stanley HE. *The Physics of Foraging: An Introduction to Random Searches and Biological Encounters*. Cambridge University Press (2011).
2. Méndez V, Campos D, Bartumeus F. Biological Searches and Random Animal Motility. In: *Stochastic Foundations in Movement Ecology: Anomalous Diffusion, Front Propagation and Random Searches* (eds Méndez V, Campos D, Bartumeus F). Springer Berlin Heidelberg (2014).
3. Shlesinger MF, Klafter J, Wong YM. Random-Walks with Infinite Spatial and Temporal Moments. *J Stat Phys* **27**, 499-512 (1982).
4. Klafter J, Blumen A, Shlesinger MF. Stochastic Pathway to Anomalous Diffusion. *Phys Rev A* **35**, 3081-3085 (1987).
5. Metzler R, Klafter J. The random walk's guide to anomalous diffusion: a fractional dynamics approach. *Phys Rep* **339**, 1-77 (2000).
6. Zaburdaev V, Denisov S, Klafter J. Levy walks. *Rev Mod Phys* **87**, 483-530 (2015).
7. Mantegna RN, Stanley HE. Stochastic-Process with Ultraslow Convergence to a Gaussian - the Truncated Levy Flight. *Physical Review Letters* **73**, 2946-2949 (1994).

(2) It is exactly the claim of Levy walks/flights that I feel most at odds with. The evidence claimed from figure 2e is in my view a massive overstatement. Even in the old days of fractals it was always clear that power-laws with less than a decade of scaling are no power-laws. Here we have 1/2 decade of proper scaling, if at all. Also in the preceding two panels c and d I would contest that the shown power-law fit makes much sense. Instead, the curves to me look much more like an exponential function or related function decaying quicker than a power-law. Having said this, I would ask the authors to study an alternative scenario that could indeed be relevant here. Namely, in Phys Rev Res 4, 033055 (2022) it was recently observed that searching birds, instead of the long-standing dogma that they perform Levy-type search processes, follow a persistent superdiffusive motion compatible with fractional Brownian motion with Hurst index larger than 1/2. To see this the authors should check how good or bad an exponential fit is to their data in figures 2c-e.

R1A2: We understand and share the concerns of Referee 1. To address these concerns, we (1) added a paragraph discussing the limitations imposed by sporozoite biology on the experimental system and analysis, (2) collected additional data from 3 independent experiments using a ~6-fold higher area of observation to improve the recording of longer tracks and ameliorate data fitting, and (3) compared exponential and power-law fit on the data of figure 2c-e as suggested by the Referee using Akaike Information Criteria to determine the best fit.

With the new dataset, AIC indicates that power-law fits our data distribution better than the exponential fit for INV and NINV parasites in St2 and St1+2, reinforcing that high-motility state parasites are moving in a Lévy-like walk.

Now the new text reads (new line 132):

« In spite of the wealth of studies employing Lévy walk models, their identification in experimental data can be a challenging task when the trajectories are two- or three-dimensional and span less than two orders of magnitude ¹⁹. Differently from species that can travel long distances ^{1, 14}, sporozoites perform a cutaneous migration limited in space and time by their biology. Accordingly, parasite speed and thus, displacement, decrease overtime as well as BV invasions, which mostly occur within 40 minutes after sporozoite inoculation into the skin ⁵. To optimize the recording of longer trajectories and improve data analysis, we tracked 74 additional sporozoites in a volume of 664 x 901 x 30 μm^3 for 40 minutes, representing a ~6-fold increase in the area of observation. To approach the problem of the three-dimensional analysis, we used protocols allowing the separation of a given path in its distinct directions, thus considerably facilitating the characterization of the movement turns ^{20, 21}. It is also known that the relative advantage of optimal Lévy searches decreases in three-dimensional search spaces due to the smaller oversampling of random walk trajectories ^{22, 23, 24, 25, 26}. Of note, in our experimental setup, while searching for a BV, sporozoites move in a nearly two-dimensional path in the upper dermis, with thickness (30 μm) considerably smaller than the tissue surface extension (320-664 μm x 320-901 μm). »

The new data is presented in the new figure 2c-f:

Figure 2. High-motility mode and Lévy walk. (a) Representative examples of INV (blue) and NINV (red) parasite tracks showing continuous segments of St2 high-motility (HM, lines) and St1 low-motility (LM, circles). Scale bar, 40 μm . (b) HM and LM state alternation over time for two INV and two NINV exemplary trajectories. (c-f) The best PL and EXP fits for (c) INV parasites in the St2 mode, (d) INV parasites combining the St1 and St2 modes, (e) NINV parasites in the St2 mode (f), NINV parasites combining the St1 and St2 modes. The insets in (d) and (f) show the fits for St1 mode alone while insets in (c) and (e) display log-normal plots of the St2 mode, helping to highlight the long-tail distribution of the experimental data. Circles show empirical data, dashed (black) and dotted-dashed (cyan) lines respectively represent the best PL and EXP fits. The adjusted parameters μ and ℓ_{min} as well as the total number of steps analyzed are indicated on each graph. Akaike Information Criterion (AIC) values and weight indicate the best fit highlighted using bold italic letters for each dataset. In the insets in (d) and (f), numerical values correspond to (μ ; ℓ_{min} ; AIC value, AIC weight). (c-f) Pooled data from seven independent experiments, yielding 27 INV tracks and 161 NINV tracks.

And the results are described as shown below (new line 155):

« A continuous power-law (PL) distribution was fitted to each group individually or to the combined St1+2 groups for INV and NINV (Fig. 2c-f, black dashed line). To test other potential mechanisms leading to super-diffusion, like fractional Brownian motion ²⁸, we additionally fitted an exponential (EXP) distribution ^{29,30} to the same groups (Fig. 2c-f, cyan dotted-dashed line). Using Akaike Information Criterion (AIC), we found that PL is comparatively the best distribution for the high-motility St2 mode, with μ INV = 1.6 and μ NINV = 2.6, which are exponents leading to a super-diffusive Lévy-like walk (Fig. 2c and 2e). Conversely, for both INV and NINV at low-motility St1 mode, step lengths are best fitted by an EXP distribution (Fig. 2d and 2f, insets). The step length range is very small for the St1 mode, which displays rather short steps, and thus has a different distribution trend compared to the St2 mode (Fig. 2d and 2f, arrows). Importantly, the optimal ℓ_{\min} fitting parameter for the St2 mode data is not substantially altered by adding the St1 mode data (Fig. 2c and 2d for INV and Fig. 2e and 2f for NINV), implying that longer steps are almost absent for sporozoites in the St1. Consequently, PL is the best fit for both INV (Fig. 2c and 2d) and NINV (Fig. 2e and 2f) parasites, which display similar super-diffusive μ exponents ($\mu < 3$) for both St2 alone and combined St1+2 distributions. »

Moreover they should check the velocity autocovariance function of the observed motion. For the superdiffusive parts relevant for figure 2e these might be positively correlated, and these correlations should fall off slowly, compare the analysis in the above paper. Given that there is active motion involved in the system such a scenario is not unlikely and could provide a similarly reasonable explanation towards optimised search.

For the subdiffusive parts reported by the authors I would also ask for an analysis of the velocity autocovariance. This might show compatibility with a viscoelastic behaviour. Alternatively, maybe trapping is relevant, i.e., more of a subdiffusive continuous time random walk scenario? Please show some more details on this part of the motion, as well.

R1A2b: We thank and agree with the Referee 1 that velocity autocovariance (or the very closely associated autocorrelation) is an important characterization for the parasite locomotion dynamics. In fact, autocovariance should be computed in the same temporal windows discussed in the manuscript, those typifying the distinct movement modes (St1 and St2) and ranges for the Hurst exponents. We performed this analysis, which is now presented in the supplementary figure 1 as shown below:

Velocity autocorrelation analysis

Considering the sporozoite velocities in the time instants t and $t-1$, with $\theta_{t,t-1}$ denoting their relative angle, we define the component $v_c(t) = |\mathbf{v}(t)| \cos[\theta_{t,t-1}]$. Then, we can calculate the velocity autocorrelation function of lag τ as ⁸ (for T the maximum time).

$$C(\tau) = \frac{\sum_{t=0}^{t=T-1-\tau} (v_c(t) - \langle v_c \rangle) (v_c(t+\tau) - \langle v_c \rangle)}{\sum_{t=0}^{t=T} (v_c(t) - \langle v_c \rangle)^2}. \quad (S1)$$

The results for INV and NINV, both in the states St1 and St2, are shown in the Supplementary Figure 1. These graphs have a temporal scale limitation due to the data processing method. Successive times t and $t-1$ represent, in fact, averages over intervals of four seconds, establishing a lower-bound for the temporal resolution. Also, autocorrelation statistics tend to be very poor for rather short temporal series, here corresponding to individuals with less than 30 steps (120 seconds) for INV and with less than 100 steps (400 seconds) for NINV. These samples were therefore eliminated from the analysis. All this, of course, hinders a high precision estimation for the auto-correlation rate decay. Furthermore, for the St1 state, the velocities and step lengths tend to be considerably smaller, conceivably making $C(\tau)$ to be more susceptible to other factors, like the dermis local structures. Even then, expected general trends can be observed in the Supplementary Figure 1. First, in all cases, after 4 s (the minimum acquisition time) the correlation decay is slow and positive up to a certain τ_{\max} . Second, positive autocorrelation during a certain time characterizes a specific movement mode: very small steps and low speed for mode St1 and larger steps and higher speeds for mode St2. For $\tau_{\max} < 20$ s for INV and $\tau_{\max} < 40$ s for NINV, the autocorrelation is positive and higher for St2 than for St1. For the state St2 (specially for the NINV) such τ_{\max} fairly corresponds to the threshold for which the Hurst exponent are higher than 0.5 (Fig. 3a).”

Supplementary Figure 1. Velocity autocorrelation. Velocity autocorrelation, Eq. (S1), for the St1 (blue) and St2 (red) modes of (a) INV and (b) NINV parasites for the lag, i.e. τ , given in seconds. Lines represent the average and whiskers the standard deviation. Pooled data from 7 independent experiments ($n=188$ tracks) yielding 10 analyzable segments for St1 INV, 15 for St2 INV, 16 for St1 NINV and 20 for St2 NINV.

SI Reference:

8. Box GEP, Jenkins GM, Reinsel GC, Ljung GM. Time series analysis: forecasting and control, Fifth edition / edn. John Wiley & Sons, Inc. (2016).

Supplementary figure 1 is now introduced in the main text in the new line 61:

“For both St1 and St2 modes, the time span for positive velocity autocorrelation have been estimated and discussed in the Supplementary Figure 1.”

Regarding the St1 mode, experimental issues associated to the acquisition process, explained above, make difficult a clear-cut identification of their general correlation features and the additional analysis asked by the Referee 1. Only general comments are made in this supplementary text.

Finally, the raised point regarding viscoelastic behavior is rather interesting, e.g., being one of the examples in the Phys. Rev. Res. paper discussing sub-diffusion for Rhodamine molecules in water film. Some researchers, such as Andy Reynolds, who is well known in the area of super-diffusion in animal movement, in particular for insects, have analyzed data showing that in certain instances, super-diffusion can be caused by the medium itself. For instance, when insects profit from wind flows in their foraging activities, being carried out by turbulent streams. But note that in both examples, the “particle” is not propelling itself. The medium is fully responsible for generating the locomotion pattern.

In our present case the parasites are indeed executing movements against a given medium, conceivably in a way to optimize the encounter rate for their goal target “objects”, blood vessels, and the finding of the pericyte-associated hotspots along such vessels. Nonetheless, a possible question is if the emerging locomotion patterns could result from the environment elastic and viscous forces influencing the animal dynamics, say, as a pure mechanical response to the medium. This does not seem to be the case. For instance, the dermis is rather heterogeneous, displaying a great diversity of acellular and cellular structures. Therefore, a same single predetermined dynamical trend (i.e., with the sporozoites not switching their locomotion modes) performed in these many different regions would lead to a much greater diversity of movement behavior than those observed. Also, the data should display specific modes of movement (say associated to sub-diffusion) only in specific parts – corresponding to determined types of tissues – of the dermis full volume analyzed. This was not identified in the data. This of course would constitute a valid subject for a future study, but we do not think it should be addressed in the present work.

(3) For the reported anomalous diffusion exponents I ask the authors to also show some examples for the mean squared displacements. Otherwise it is quite difficult to assess the relevance of the stated exponents. In particular it would be nice to show both ensemble and time averaged MSDs and discuss their statistics.

R1A3: We performed the requested analysis and discussed their statistics in the novel supplementary figure 4-7 as shown below:

Extra analysis for certain trajectories-related quantities

Supplementary Figure 4. Examples of traveling distances and MSD from the origin. (a-b) The path length, i.e. the total traveled distance, as function of time for two distinct parasites, one INV and one NINV. (c-d) For these same trajectories, the MSD versus time computed in reference to the tracks starting points at $t=0$. (e) MSD of NINV versus time in log-log scale corresponding to the data depicted in figure 3b of the main text. Here, the distinct MSD curves are clustered (c1, c2 and c3) according to the kmeans protocol, selected by their slope values (m).

The actual trajectory shapes can change considerably for the different individuals (Fig. 1a). Likewise, the MSD, if not taken in time windows (refer to the Hurst exponent section in the Materials and Methods) tends to present strong variability, complicating statistical analysis. As illustrative examples, we show in the Supplementary Figure 4, the total traveled distance, i.e. the path length, as function of time for two specific parasites, one INV and one NINV. For these tracks we also display the MSD versus time (always calculated in relation to the starting point at $t=0$). Observe the distinct length scales in these two cases. Here, the INV parasite finally invade BV at the instant $t=804$ s. As discussed in the main text, for the NINV population, the k-means clustering of the MSD slopes (m) separate the Hurst exponents into three groups, c1, c2 and c3 (Fig. 3b). Such separation can be further understood from the Supplementary Fig. 4e, where we clearly see the overall trends, $m > 1$ for $H > 1/2$ (super-diffusion), $m=1$ for $H=0.5$ (normal diffusion) and $m < 1$ for $H < 0.5$ (sub-diffusion).

As discussed in the main text, the Hidden Markov Model (HMM) identifies along the distinct tracks the trajectory stretches corresponding either to the St1 (low-motility) or to the St2 (high-motility) movement modes. A relevant information is thus the distribution of both size and duration of these St1 and St2 stretches modes, providing a general overview on the spatial-temporal ranges for the exploration/exploitation behaviors. This is shown in Supplementary Figure 5, which includes all the parasite tracks.

Supplementary Figure 5. Spatial and temporal distribution of the St1 and St2 movement modes. Distribution of the stretch lengths (in the inset, stretch times) for the St1 and St2 locomotion modes as identified by HMM analysis. Here the full set of tracks for all parasites are considered (7 independent experiments, 188 tracks). As expected, the stretch lengths tend to be much longer for the high-motility St2 movement behavior.

Finally, considering the entire collection of parasite tracks and four distinct time windows τ , we have obtained the corresponding MSD along each one of the stretches classified as St2 mode via the HMM. Then, we have generated the distributions of MSD lengths shown in the Supplementary Figure 6. The same procedure for the St1 has led to the distributions in the Supplementary Figure 7. We clearly see that the length sizes are much longer for the St2 mode. Moreover, in this case the distributions are much broader, presenting fatter tails.

Supplementary Figure 6. Distribution of the MSD in different time windows of analysis for the St2 mode. For four time window values, the corresponding MSD distributions resulting from all the stretches of the St2 mode identified by the HMM were computed for the entire set of parasites tracks (7 independent experiments, 188 tracks).

Supplementary Figure 7. Distribution of the MSD in different time windows of analysis for the St1 mode. For four time window values, the corresponding MSD distributions resulting from all the stretches of the St1 mode identified by the HMM were computed for the entire set of parasites tracks (7 independent experiments, 188 tracks).

(4) I did not understand the statement at the beginning of the paragraph starting in line 52. Does it mean that parts of individual trajectories were discarded? If yes, this could alter the correlations of the associated time series. Please explain and rephrase.

R1A4: We thank the Referee 1 for bringing this point to our attention. Our apologies for not being clear in the text. In the old manuscript version, the HMM analysis was done excluding the points where the parasite was not moving to improve the detection of the low motility state in detriment of the immotile state (old Fig. 1). Of note, Hurst exponent analysis was performed without this velocity filter.

To avoid confusion, in the new version of the manuscript, we removed the velocity filter to perform the HMM analysis. Despite decreasing the frequency of St1 points in the trajectories, the new figure 1 shows the same essential messages that we had previously shown in the old figure 1.

The new figure 1 and legend are shown below:

Figure 1. Hidden Markov Model (HMM) analysis of sporozoite motility. (a) Typical parasite tracks (scale bar: 40 μm) showing state 1 motility (St1, circles) of BV invaders (INV, blue) and non-invaders (NINV, red) strikingly co-localizing in the same BV areas. HMM state 2: gray lines, BV invasion sites (BVI): black crosses. **(b)** Quantification of sporozoite velocity showing low-speed St1 and high-speed St2 motility. **(c)** Frequency of sporozoite turning angle

distribution. **(d)** Distance to the nearest BV according to HMM states and invasive phenotype. **(e)** INV/NINV HMM states distribution (INV=2,819; NINV=11,581; Two-tailed Fisher's exact test $P < 0.0001$). **(f)** State frequencies over time for twenty-four INV/NINV sporozoites **(g)** Example of successive INV St1 interactions. St1: circles, St2: gray line, BVI: black cross and dashed arrow. Upper graph: state alternation over time. Image: INV example showing an extravascular St1 stretch (blue circles) and multiple St1 stretches (\geq quadruple) on an invaded vessel. **(h)** Number of INV displaying St1-stretches close (invaded vessel) or far (other vessel) from BVI, or within the extravascular tissue (no vessel). **(i)** Frequency distribution of distance between the last INV St1 and BVI. For parasites flushed with the bloodstream after the last St1, the distance was considered as zero. **(b)** and **(d)** Multi-comparison using Kruskal-Wallis/Dunn's test. Lines denote the median, crosses the mean and whiskers indicate the 10th and 90th percentiles. **(b-i)** Data from four independent experiments.

(5) There are many statements about how Levy search can improve the search efficiency in literature. However, this is quite dimension-dependent. In one and two dimensions, there is a clear advantage, however, in three dimensions this is no longer significant, due to the much smaller oversampling of random walk processes in higher dimensions (Phys Rev Lett 124, 080601 (2020)). In one dimension and effectively one dimension (see, e.g., Phys Rev Lett 95, 260603 (2005)) the best-case corresponds to a Levy index of 1, i.e., $\mu=2$ in the present notation. However, small disturbances can significantly reduce this advantage, as analysed in Proc Natl Acad Sci 111, 2931 (2014). These points should be mentioned in the discussion.

R1A5: As already mentioned in R1A2, please find below the new sentences addressing this point with 5 new references (Refs 21-25):

“It is also known that the relative advantage of optimal Lévy searches decreases in three-dimensional search spaces due to the smaller oversampling of random walk trajectories^{21, 22, 23, 24, 25}. Of note, in our experimental setup, while searching for a BV, sporozoites move in a nearly two-dimensional path in the upper dermis, with thickness (30 μm) considerably smaller than the tissue surface extension (320-664 μm x 320-901 μm).”

(6) There exist examples for Levy walks in cellular systems involving molecular motor motion. These should be included in the discussion: Nature Comm 9, 344 (2018); Nature Mater 14, 589 (2015).

(7) Change point analysis was shown to work nicely in machine learning approaches. The latter have also become very successful in identifying Levy walk motion. Maybe they could be used to analyse the current data. While I do not insist that the authors study this in the framework of the current manuscript, the potential should be mentioned in the discussion, compare Nature Comm 12, 6253 (2021); J Phys A 55, 244005 (2022); Nature Comm 13, 6717 (2022).

R1A6 and A7: We thank the Referee 1 for this suggestion. We added a new paragraph in the main text to take in consideration the points of comments 6 and 7. The new text is shown below (new line 171):

“Interestingly, the dual motility behavior of sporozoites - alternating low (St1) and high (St2) motility modes and resulting globally in a super-diffusive Lévy-like walk – displays some similarities with other biological processes. For instance, the transport of ribonucleoproteins inside neurons also displays two alternating movement states, a very slow and a ballistic mode of transport. Their combination can also result in a super-diffusive Lévy-like walk by convoluting the time duration of each state³¹. An important future investigation should be to determine which molecular mechanisms could account for the St1 and St2 transition in the sporozoite search strategy. A study of intracellular cargo transport has shown that Brownian-like steps allied to self-organizing directional reinforcement may give rise to Lévy walks with $\mu < 3$ ³². Although the angle distribution for sporozoites moving in St2 mode might be an indication of directional reinforcement (Fig. 1c, high frequency of turning angles = 0 rad, characteristic of parasites moving in consecutive steps without direction change), whether this mechanism is involved in the super-diffusive Lévy-like walk of sporozoites still needs to be determined. Finally, from a more technical point of view, recently machine-learning methods have been successfully used to pinpoint Lévy walks and other anomalous diffusive modes in distinct datasets. They likewise might be helpful to study potential changes of diffusive modes within parasite trajectories^{33,34,35}. »

(8) In figure 1, could you replace the green crosses by some darker orange?
For readers with colour vision problems the green crosses are virtually invisible.

R1A8: Our apologies for that. The green crosses were substituted by black crosses in the new figure 1 as seen in R1A4.

Reviewer #2 (Remarks to the Author):

The new findings of this paper are interesting and an important contribution to the field. Specifically they show that sporozoites, once close to blood vessels, perform a local exploration to detect blood vessel entry points that are scattered within a restricted area characterized by the presence of pericytes and blood vessel junctions. These data, which appear to be well-supported, bring us a little closer to understanding how sporozoites search for and recognize blood vessel entry points. However, there are two major issues concerning their findings around the two types of motility exhibited by parasites that need to be addressed:

We would like to thank the Referee 2 for the interest and the positive comments about our work, and for raising points that allowed us to clarify and to improve the manuscript.

1) Blood vessels in the dermis, which is the tissue where the majority of mosquito probing occurs, are not rare. Indeed capillary density in the dermis is between 50 to 70 capillaries per mm². Given that sporozoites are 10 microns in length, this means that there is a blood vessel within a few sporozoite lengths in any given part of the dermis. Thus, a Levy-walk does not make sense as blood vessels are abundant. What is not abundant are the entry points and I wonder if they are confusing the Levy-like searching that occurs once a parasite has encountered a blood vessel with a Levy-walk that is frequently used by some organisms to find a rare item. The references they use to support that blood vessels are rare in the dermis are not focused on blood vessel density in the dermis and any image of labeled blood vessels in the skin can immediately dispel the notion that they are rare.

R2A1. These are indeed very pertinent comments. Despite this density of capillaries, Blood Vessels (BV) represent only a small fraction of the total skin volume. Skin blood and vessel volume are respectively estimated to be around 0.2-5% by tissue spectroscopy (Jacques et al, 1996 – new ref 6) and 2-5% by optoacoustic mesoscopy imaging (Berezhnoi et al, 2017 – new ref 7). This is compatible with the value of 7.2 ± 0.3 % obtained in our set-up (new supplementary figure 3). Of note, sporozoites invade vessels with relatively small diameter (mean= 11.9 ± 3.6 μm), which represent only a fraction of the total BV volume. We introduced this quantitative information in the main text (new line 33), adding the two new references (6 and 7) at the end of the sentence that now reads:

« For instance, to disseminate and infect the liver, *Plasmodium* sporozoites deposited in the extravascular regions of the skin first need to find and invade blood vessels (BV)^{4,5}, which only represent approximately 5% of the dermis volume^{6,7}. »

In agreement with Referee 2 observation, we removed the sentence referring to the BV density and changed it by the fractional volume analysis of dermal microvessels, presented in the new supplementary figure 3. We also removed “rare BV” from the main text, which is a vague notion, and replaced it by “intravasation sites”. The new text now reads (new line 127):

“Owing to the low fractional volume of the microvasculature in our experimental setup (7.2 ± 0.3 %, Supplementary Fig. 3), the relative low rate of BV invasion, and the plasticity of Lévy walks, considered as one of the simplest mechanisms allowing behavioral adjustments as response to the environment ¹⁸, we hypothesized that parasites in high-motility mode perform a super-diffusive Lévy walk to optimize the finding of intravasation sites.”

Our data show that the non-super-diffusive St1 movement mode is mainly performed around or along BV, associated with an exploitation type of locomotion to find hotspots of BV invasion (Fig. 1). On the other hand, the super-diffusive Lévy-like locomotion, represented by the St2 motility mode usually takes place between successive BV interactions. Taking in consideration the 3D volumetric analysis of the microvasculature, a Lévy-like search strategy to locate BV does make sense. Sometimes sporozoites are close, but don't readily interact with BV, and can even move away from the vasculature. In fact, motile NINVs require in median as long as ~ 100 s to encounter a BV. Interestingly, NINVs were found to contact BVs in average 23 ± 10 μm away from the nearest pericyte, which is not significantly different from the half-distance between pericytes (27 ± 12 μm) and indicates that super-diffusive locomotion enables sporozoites to encounter random regions of BV. Altogether, this suggests that Lévy-like motility likely optimizes the finding of BV in general, while subsequent vessel exploitation enables sporozoites to probe for intravasation hotspots.

2) A previous study that they cite, (Hopp et al., Longitudinal analysis of Plasmodium sporozoite motility in the dermis reveals component of blood vessel recognition. eLife 2015), uses quantitative motility analysis to demonstrate what had been observed in one of this group's previous papers (Amino et al., Quantitative imaging of Plasmodium transmission from mosquito to mammal. Nat Med 2006), namely that sporozoites display two types of motility in the skin: A high speed, more linear motility that maximizes tissue exploration and a low-speed more constrained motility once they are in the vicinity of blood vessels which is likely related to blood vessel recognition. Since it is important that results are reproducible, it is not a problem that this group, using different methodologies, demonstrate the same thing. However, it needs to be clearly stated that these two types of motility, with one being associated with blood vessels, has been previously shown.

R2A2: We completely agree with the Referee 2 comment. As such, we clearly stated this point in two sentences, one in the introduction (line 31 old version) and one in the beginning of figure 2 description (line 40 old version).

“After inoculation into the skin, sporozoites are known to engage in a non-Brownian movement ⁶, decreasing their speed upon interaction with BV ^{5,6},...”

Since sporozoites present behavior alternation, switching between a high-speed avascular motility and a slower and more constrained motility at the vicinity of BV ^{5,6} ...”

We added a third sentence in the new version of the manuscript to stress this point (new line 64) :

“These quantifications are in good agreement with previous observations of avascular high-motility and vascular low-motility migration patterns of sporozoites in the dermis ^{5,8}.”

Reviewer #3 (Remarks to the Author):

The inoculation of sporozoites into the human host is an absolutely critical stage of the parasite lifecycle, both as the most significant bottle neck in the mammalian host and a vital target of current malaria vaccines. The fleeting nature of this step means that we still understand little of the parasite dynamics/behaviours underpinning these early steps. This work applies intravital imaging approaches with innovative statistical models to understand sporozoite search behaviour. This adds significant insight into how sporozoites are able to both find blood vessels and what conditions are required for intravasation. I think this would be of significant interest to the broad audience of Nature Comms, although would benefit from some greater explanatory detail to ensure it is as accessible to the broad scientific community as possible. The manuscript is well written with clear concise intro and discussion sections, but I found it a bit hard to follow through some of the results sections. Whilst this may be my shortcoming, given the interdisciplinary nature of work a few sections could do with some additional more basic explanation to make sure it is as accessible as possible and ensure the exciting key findings hit home. I will give a few examples below, but worth authors going through the manuscript themselves with an eye for simplifying technical language (without sacrificing technical content). I would also be keen to see slightly expanded explanation of how detection of a BV affects followed.

We would like to thank the Referee 3 for the compliments and the pertinent comments, which greatly helped us to improve our manuscript.

Major points

1) Line 85 – The sentence here indicates that after a failed St1 stretch the sporozoite reverts back to search mode. This didn't make sense to me, as surely once a vessel is found it makes sense to just try somewhere further along the same vessel. That does seem to be what happens for example Fig 1G. Does initial contact then lead to confinement of the search pattern in proximity to the BV?

R3A1: Thanks for the question. This sentence refers to the figure 1f (NINV) which represents a typical NINV behavior (the graph of the old version represented the state frequency of 4 sporozoites, in the revised manuscript it represents 24 sporozoites). In this figure the values denote the frequency of states over periods of 100s, thus individual St1 stretches are averaged with St2 stretches during this time frame. We added this precision in the modified version of the manuscript.

Please note that it is after an unsuccessful intravasation (BV invasion), not after a failed St1 stretch that NINV return to a high motility state (transition between the second and third blue rectangle in NINV – fig 1f). The new text now reads (line 86):

« Altogether, the HMM analysis shows that the sequence of NINV states is compatible with a predominant high-motility search mode (Fig. 1f, NINV, $St2\% > St1\%$, first 200s), followed by an increase in $St1$ behavior upon BV encounter or extravascular interaction. After an unsuccessful intravasation, NINV return to a high-motility search mode such as in a trial-and-error approach. The sequence of INV states is compatible with an initial high-motility general search mode similar to NINV behavior (Fig. 1f, INV, $St2\% > St1\%$, first 200 s). However, upon BV encounter, INV switch states and perform a low-motility, intermittent, local vessel scan ($St1\% > St2\%$), which ultimately results in a BV invasion. »

Fig.1f:

2) Fig 1A Similarly there looks to be an interesting ST1 hotspot just down and left from the centre site in fig 1. This looks like it could be on a BV but it looks slightly larger and less defined. Is this also a vessel? Is it different in type to the others (given its twice diameter). This cluster doesn't have an entry point. Are there other such examples?

R3A2: Referee 3 is right. It is a vessel, probably a post-capillary venule given its diameter and it is less defined because it is deeper. Despite a "St1 cluster" of INV and NINV sporozoites in this vessel there is no intravasation site. This is an example of INV depicted in the graph of the figure 1g (other), ie, parasites displaying $St1$ stretches on vessels other than the invaded one. The quantification of this type of INV behavior is shown in the figure 1g (other).

3) Fig 4 was there any analysis done to look at the NINV parasites. It would be interesting to establish whether they just failed to find the right (pericyte proximal) place to invade, or whether they were intrinsically less fit for invasion etc. Instead of using invasion site, you could use the $St1$ motility mode as a marker to examine whether $St1$ mode in either NINV or INV correlates with pericyte position. This could be informative and presumably existing experiments could be reanalysed in this way?

R3A3: This is an excellent point. As shown in figure 1a (zoomed below) NINV can perform $St1$ stretches in very close proximity to a hotspot of invasion (characterized by two adjacent black crosses) indicating that finding and moving in $St1$ mode in this area of invasion is a common characteristic for both INV and NINV.

Although, St1 motility can sometimes coincide with the presence of a pericyte body, it is not an exclusive marker of sporozoite-pericyte interaction (please see the pictures below, red circles for St1 NINV and blue circles for St1 INV). We are currently performing experiments to better understand parasite and host factors required for successful intravasation, including the mechanisms of BV invasion. This data will be submitted in another manuscript focused on this subject.

4) Technical language – I found a few sections hard to follow, particularly sections like lines 94-99 are quite hard to follow. Phrases like ballistic behaviour are no doubt right but not super obvious in meaning to me. A lay explanation of a levy walk would be useful too (I googled it).

R3A4: We thank the Referee 3 for bringing this point to our attention. We added more detailed explanation about technical terms (maximum length L , ballistic) in the main text, and complemented this information with a supplementary figure to illustrate some of these terms, and the relationship between the PL exponent μ and the process of diffusion which this exponent characterizes. The new explanations in the main text and supplementary figure are shown below (new line 100):

“To a great extent ¹⁷, for a large enough maximum length L , e.g., usually the largest linear length in the searching region, the essential aspects of Lévy walks can be described by a truncated power-law distribution of the step length ℓ : $P(\ell) \sim C/\ell^\mu$ for $\ell_{\min} \leq \ell \leq L$, with ℓ_{\min} a lower cutoff and C a proper normalization constant ¹. Decreasing μ values within the interval $1 < \mu < 3$ translate into an increasing degree of super-diffusiveness. The limit $\mu \rightarrow 1$ leads to ballistic behavior consisting essentially of extremely large consecutive concatenated steps, while $\mu \geq 3$ corresponds to typical normal diffusion, characteristic of Brownian-like motion ¹ (Supplementary Fig. 2).”

Supplementary Figure 2. Relationship between diffusive behavior and μ_{PL} . Lévy walk simulations⁹ illustrating μ_{PL} (bottom of each simulation) and its relationship with (a) ballistic behavior (limit $\mu_{PL} \rightarrow 1$), (b) super-diffusive behaviors ($1 < \mu_{PL} < 3$), and (c) normal diffusion ($\mu_{PL} \geq 3.0$). Panels highlight how smaller μ values lead to larger steps and enhanced diffusion, while for larger μ values ($\mu=3$), smaller steps dominate the motion leading to normal diffusion.

Also, a more general description about Lévy walks (as well as about Lévy flights, sometimes confused with Lévy walks) is given in the new SI. Although such extended discussion requires some mathematical definitions, the less restricted space allowed by the SI permitted us also to mention some qualitative facts about Lévy walks strategies.

5) General – Whilst it makes sense to not change formatting between submissions, I think the final version should be divided up into results subheadings. This would help clarify the messages of each section. I also don't see any movie attachments in the version I have. I think adding some of the example images as supplementary video files would really help to visualise the distinct patterns (perhaps with some annotation – for example video for fig 1g with marked phases would be great.

R3A5: We thank the Referee 3 for the suggestions. The manuscript does not have subheadings because it was written in a letter format. If the editor agrees we don't see any problem to include them in the final version. We also added supplementary movies showing examples of INV and NINV search behavior as well as hotspots of invasion associated with pericytes.

Minor points

L142 – inferior could replaced with less than?
It is done!

Fig 3 D colour of C3 legend key is pink instead of orange.
It was changed!

Fig 3 legend “Symbols represent the average of 4 independent experiments” could you

define what is meant by independent experiments here or in methods (i.e. is it different mice, different mosquito infections or different videos etc etc.)

This information is included now in the “Reporting Summary” for all experiments.

“One independent experiment refers to a 40-minute long longitudinal recording of freshly dissected salivary gland sporozoites micro-injected into one mouse ear. All intravital imaging experiments presented in the manuscript were repeated at least 4 times in independent animals and using sporozoites from at least 4 independent mosquito infections. Reported findings were replicated across all replicates with similar trends and data were pooled for statistical analysis. Specifically, analysis of sporozoite motility and search strategy (Fig. 1-3) was performed on tracks of 108-188 sporozoites collected over 4-7 independent experiments (4-6 mice, 6 mosquito infections).”

Reviewer comments, second round -

Reviewer #1 (Remarks to the Author):

I appreciate the thorough revision of the manuscript and agree with the points brought forth in the rebuttal. After clarification of my concerns I very warmly recommend publication of this surely influential paper.

Reviewer #2 (Remarks to the Author):

This very nice study is now presented with more clarity and I think is much improved. The authors have addressed my concerns and I fully support its publication.

There are 2 statements that are somewhat misleading - its not necessary but if possible to change them, it would be more accurate:

1. Line 138 is probably only true for *P. berghei*. Experiments with the other rodent parasite *P. yoelii* suggests their motility time in the skin is of longer duration and a significant number of blood vessel entry events have been observed at 2 hrs (Hopp et al., EMBO Mol Med 2021).

2. The last sentence of the Discussion (lines 289-292) is confusing and perhaps not accurate. Stellate cells are in the Space of Disse and thus not accessible to sporozoites until they exit the hepatic sinusoid. Thus, in the skin the sporozoites are interacting with the ab-luminal side of the vessel while in the liver they are interacting with the luminal side. In the paper that they cite, the authors suggest that perhaps the highly sulfated heparan sulfate proteoglycans made by stellate cells protrude through the fenestrae - but the cells themselves are not exposed in the sinusoid so its not clear how this relates to the very interesting findings in this paper on pericytes found surrounding the parts of the dermal vessels.

Reviewer #3 (Remarks to the Author):

The authors have responded with a detailed and comprehensive rebuttal, resulting in significant improvements to the manuscript. I can confirm my comments are fully addressed in this new version.

POINT-BY-POINT RESPONSE TO THE REVIEWERS' COMMENTS

We thank [redacted] the Reviewers for the constructive comments, enthusiasm, and support.

Reviewer #1 (Remarks to the Author):

I appreciate the thorough revision of the manuscript and agree with the points brought forth in the rebuttal. After clarification of my concerns I very warmly recommend publication of this surely influential paper.

Reviewer #2 (Remarks to the Author):

This very nice study is now presented with more clarity and I think is much improved. The authors have addressed my concerns and I fully support its publication.

There are 2 statements that are somewhat misleading - its not necessary but if possible to change them, it would be more accurate:

1. Line 138 is probably only true for *P. berghei*. Experiments with the other rodent parasite *P. yoelii* suggests their motility time in the skin is of longer duration and a significant number of blood vessel entry events have been observed at 2 hrs (Hopp et al., EMBO Mol Med 2021).

We thank again reviewer 2 for the comment.

Even though *P. yoelii* sporozoites present higher number of blood vessel entry events between 60-120 min than *P. berghei* and *P. falciparum* (Hopp et al, 2021, Fig. 6, below), the graphs show that the percentage of invasion at earlier timepoints (5 to 20 min) in 4 min-movies analysis is higher than at later time points (60-120 min), and thus most of invasions will likely occur during the first hour post-inoculation.

This result is in accordance with unpublished results from our lab where in most experiments with varying number of BV invasion events, the majority of Py intravasation occurred before 45 min. However, in one experiment (green triangles), the kinetics of intravasation was delayed, by an unknown reason.

However, since it is not possible to accurately calculate the intravasation kinetics from Hopp et al (Fig.6) and our results are unpublished, we specified in the sentence that we are referring to *P.berghei*. Now the new text reads:

“Accordingly, parasite speed and thus, displacement, decrease overtime as well as BV invasions, which mostly occur within 40 minutes after *P. berghei* sporozoite inoculation into the skin ⁵.”

2. The last sentence of the Discussion (lines 289-292) is confusing and perhaps not accurate. Stellate cells are in the Space of Disse and thus not accessible to sporozoites until they exit the hepatic sinusoid. Thus, in the skin the sporozoites are interacting with the ab-luminal side of the vessel while in the liver they are interacting with the luminal side. In the paper that they cite, the authors suggest that perhaps the highly sulfated heparan sulfate proteoglycans made by stellate cells protrude through the fenestrae - but the cells themselves are not exposed in the sinusoid so its not clear how this relates to the very interesting findings in this paper on pericytes found surrounding the parts of the dermal vessels.

Indeed, the sentence was confusing, our apologies for the lack of intelligibility. We have modified the main text to specify that the binding of sporozoites to molecules on the pericyte membrane or to extracellular matrix molecules secreted by pericytes could occur both on the abluminal side of the cutaneous blood vessels or in the luminal side of hepatic sinusoids via endothelial fenestrations, as suggested in the ref 43 and schematized by Prudencio et al, Nat Rev Microb 2006 (below).

Copyright © 2006 Nature Publishing Group
Nature Reviews | Microbiology

The modified text is shown below:

Interestingly, based on *in vitro* binding experiments, hepatic pericytes, known as stellate cells, have been proposed to play a role in the recognition and arrest of blood circulating sporozoites in the liver sinusoids ⁴³. Molecular interactions between sporozoite surface adhesins and pericyte molecules have also been identified ^{43,44}, hinting at the specific binding of sporozoites to pericyte-secreted extracellular matrix or surface molecules exposed either to the blood circulation via the fenestrated liver sinusoids or on the abluminal side of cutaneous BV. These specific interactions could be involved in the switch of sporozoite motility behavior at the endothelial barrier interface allowing the successful crossing of these cellular barriers in the skin and liver.

Reviewer #3 (Remarks to the Author):

The authors have responded with a detailed and comprehensive rebuttal, resulting in significant improvements to the manuscript. I can confirm my comments are fully addressed in this new version